# Counterfactual Evaluation of Peer-Review Assignment Policies

**Martin Saveski**
University of Washington
msaveski@uw.edu

**Steven Jecmen**
Carnegie Mellon University
sjecmen@cs.cmu.edu

**Nihar B. Shah**
Carnegie Mellon University
nihars@cs.cmu.edu

**Johan Ugander**
Stanford University
jugander@stanford.edu

## Abstract

Peer review assignment algorithms aim to match research papers to suitable expert reviewers, working to maximize the quality of the resulting reviews. A key challenge in designing effective assignment policies is evaluating how changes to the assignment algorithm map to changes in review quality. In this work, we leverage recently proposed policies that introduce randomness in peer-review assignment—in order to mitigate fraud—as a valuable opportunity to evaluate counterfactual assignment policies. Specifically, we exploit how such randomized assignments provide a positive probability of observing the reviews of many assignment policies of interest. To address challenges in applying standard off-policy evaluation methods, such as violations of positivity, we introduce novel methods for partial identification based on monotonicity and Lipschitz smoothness assumptions for the mapping between reviewer-paper covariates and outcomes. We apply our methods to peer-review data from two computer science venues: the TPDP'21 workshop (95 papers and 35 reviewers) and the AAAI'22 conference (8,450 papers and 3,145 reviewers). We consider estimates of (*i*) the effect on review quality when changing weights in the assignment algorithm, e.g., weighting reviewers' bids vs. textual similarity (between the review's past papers and the submission), and (*ii*) the "cost of randomization", capturing the difference in expected quality between the perturbed and unperturbed optimal match. We find that placing higher weight on text similarity results in higher review quality and that introducing randomization in the reviewer-paper assignment only marginally reduces the review quality. Our methods for partial identification may be of independent interest, while our off-policy approach can likely find use in evaluating a broad class of algorithmic matching systems.

## 1 Introduction

The assignment of papers to reviewers is one of the most important parts of the peer-reviewed publication process [1–3]. In computer science, conferences are the primary terminal venue for publications, with recent iterations of large conferences such as NeurIPS and AAAI receiving several thousand submissions [4]. As a result, these and other large conferences must rely on automated systems to decide what members of the impaneled reviewer pool will review each paper. Review matching systems typically use three sources of information: (*i*) bids, i.e., reviewers' self-reported preferences to review the papers; (*ii*) text similarity between the paper and the reviewer's publications; and (*iii*) reviewer- and author-selected subject areas. Given a prescribed way to combine these

37th Conference on Neural Information Processing Systems (NeurIPS 2023).

signals into a single score, an optimization procedure then proposes a reviewer-paper assignment that maximizes the sum of the scores of the assigned pairs [5].

The design of peer-review systems has received considerable research attention [4, 6–8]. Popular peer-review platforms such as OpenReview and Microsoft CMT offer many features that conference organizers can use to assign reviewers. However, it has been persistently challenging to evaluate how changes to peer-review assignment algorithms affect review quality. An implicit assumption underlying such approaches is that review quality is an increasing function of bid enthusiasm, text similarity, and subject area match, but how to combine these signals into a score is approached via heuristics. Researchers typically observe only the reviews actually assigned by the algorithm and have no way of measuring the quality of reviews under an assignment generated by an alternative algorithm.

One approach to comparing different assignment policies is running A/B tests. Several conferences (NeurIPS'14 [9, 10], WSDM'17 [11], ICML'20 [12], and NeurIPS'21 [13]) have run A/B tests to evaluate various aspects of their review process, such as differences between single- vs. double-blind review. However, such experiments are extremely costly in the peer review context, with the NeurIPS experiments requiring a significant number of additional reviews, overloading already strained peer review systems. Moreover, A/B tests typically compare only a handful of design decisions, while assignment algorithms typically require making many such decisions (Section 2).

In this work, we propose off-policy evaluation as a less costly alternative that exploits existing randomness to enable the comparison of many alternative policies. Our proposed technique "harvests" [14] the randomness introduced in peer-review assignments generated by recently-adopted techniques that counteract fraud in peer review. In recent years, in-depth investigations have uncovered evidence of rings of colluding reviewers in a few computer science conferences [15]. These reviewers conspire to manipulate the paper assignment in order to give positive reviews to the papers of co-conspirators. To mitigate this kind of collusion, conference organizers have adopted various techniques, including a recently introduced randomized assignment algorithm [16]. This algorithm limits the maximum probability (a parameter set by the organizers) of any reviewer getting assigned any particular paper. This randomization thus limits the expected rewards of reviewer collusion at the cost of some reduction in the expected sum-of-similarities objective, and has been implemented in OpenReview since 2021 and used by several conferences, including AAAI'22 and AAAI'23.

The key insight of the present work is that under this randomized assignment policy, a range of reviewer-paper pairs other than the exactly optimal assignment become probable to observe. We can then adapt the tools of off-policy evaluation and importance sampling to evaluate the quality of many alternative policies. A major challenge, however, is that off-policy evaluation assumes overlap between the on-policy and the off-policy, i.e., that each reviewer-paper assignment that has a positive probability under the off-policy also had a positive probability under the on-policy. In practice, positivity violations are inevitable even when the maximum probability of assigning any reviewer-paper pair is low enough to induce significant randomization, especially as we are interested in evaluating a wide range of design choices of the assignment policy. To address this challenge, we build on existing literature for partial identification and propose methods that bound the off-policy estimates while making weak assumptions on how positivity violations arise.

More specifically, we propose two approaches for analysis that rely on different assumptions on the mapping between the covariates (e.g., bid, text similarity, subject area match) and the outcome (e.g., review quality) of the reviewer-paper pairs. First, we assume *monotonicity* in the covariates-outcome mapping. Understood intuitively, this assumption states that if reviewer-paper pair $i$ has higher or equal bid, text similarity, and subject area match than a reviewer-paper pair $j$, then we assume that the quality of the review for pair $i$ is higher or equal to the review for pair $j$. Alternatively, we assume *Lipschitz smoothness* in the covariate-outcome mapping. Intuitively, this assumption captures the idea that two reviewer-paper pairs that have similar bids, text similarity, and subject area match, should result in a similar review quality. We find that this Lipschitz assumption naturally generalizes so-called *Manski bounds* [17], the partial identification strategy that assumes only bounded outcomes.

We apply our methods to data collected by two computer science venues that used the recently-introduced randomized assignment policy: the 2021 Workshop on Theory and Practice of Differential Privacy (*TPDP*) with 95 papers and 35 reviewers, and the 2022 AAAI Conference on Advancement in Artificial Intelligence (*AAAI*) with 8,450 papers and 3,145 reviewers. We evaluate two design choices: (*i*) how varying the weights of the bids vs. text similarity vs. subject area match (latter available only in AAAI) affects the overall quality of the reviews, and (*ii*) the "cost of randomization",

i.e., how much the review quality decreased as a result of introducing randomness in the assignment. As our measure of assignment quality, we consider the expertise and confidence reported by the reviewers for their assigned papers. We find that our proposed methods for partial identification assuming monotonicity and Lipschitz smoothness significantly reduce the bounds of the estimated review quality off-policy, leading to more informative results. Substantively, we find that placing a larger weight on text similarity results in higher review quality, and that introducing randomization in the assignment leads to a very small reduction in review quality.

Beyond our contributions to the design and study of peer review systems, the methods proposed in this work should also apply to other matching systems such as recommendation systems [18–20], advertising [21], and ride-sharing assignment systems [22]. Further, our contributions to off-policy evaluation under partial identification should be of independent interest.

Our code is available at: `https://github.com/msaveski/counterfactual-peer-review`.

## 2 Preliminaries

We start by reviewing the fundamentals of peer-review assignment algorithms.

**Reviewer-Paper Similarity.** Consider a peer review scenario with a set of reviewers $\mathcal{R}$ and a set of papers $\mathcal{P}$. Standard assignment algorithms for large-scale peer review rely on "similarity scores" for every reviewer-paper pair $i = (r, p) \in \mathcal{R} \times \mathcal{P}$, representing the assumed quality of review by that reviewer for that paper. These scores $S_i$, typically non-negative real values, are commonly computed from a combination of up to three sources of information: (1) text-similarity scores $T_i$ between each paper and reviewer's past work, using various techniques [23–28]; (2) overlap $K_i$ between the self-nominated subject areas selected by each reviewer and each paper's authors; and (3) reviewer-provided "bids" $B_i$ on each paper. Without any principled methodology for evaluating the choice of similarity score, conference organizers manually select a parametric functional form and choose parameter values by spot-checking a few reviewer-paper assignments. For example, a simple similarity function is a convex combination of the component scores: $S_i = w_{\text{text}} T_i + (1 - w_{\text{text}}) B_i$. Conferences have also used more complex non-linear functions: NeurIPS'16 [29] used the functional form $S_i = (0.5 T_i + 0.5 K_i) 2^{B_i}$, while AAAI'21 [6] used $S_i = (0.5 T_i + 0.5 K_i)^{1/B_i}$. The range of possible functional forms results in a wide design space, which we explore in this work.

**Deterministic Assignment.** Let $Z \in \{0, 1\}^{|\mathcal{R}| \times |\mathcal{P}|}$ be an assignment matrix where $Z_i$ denotes whether the reviewer-paper pair $i$ was assigned or not. Given a matrix of reviewer-paper similarity scores $S \in \mathbb{R}_{\geq 0}^{|\mathcal{R}| \times |\mathcal{P}|}$, a standard objective is to find an assignment of reviewers to papers that maximizes the sum of similarities of the assigned pairs, subject to constraints that each paper is assigned to an appropriate number of reviewers, each reviewer is assigned no more than a maximum number of papers, and conflicts of interest are respected [5, 27, 30–34]. This optimization problem can be formulated as a linear program. We provide a detailed formulation in Appendix A. While other objective functions have been proposed [35–37], here we focus on the sum-of-similarities objective.

**Randomized Assignment.** As one approach to strategyproofness, Jecmen et al. [16] introduce the idea of using randomization to prevent colluding reviewers and authors from being able to guarantee their assignments. Specifically, the program chairs first choose a parameter $q \in [0, 1]$. Then, the algorithm computes a randomized paper assignment, where the marginal probability $P(Z_i = 1)$ of assigning any reviewer-paper pair $i$ is at most $q$. These marginal probabilities are determined by a linear program, which maximizes the expected similarity of the assignment subject to the probability limit $q$ (detailed formulation in Appendix A). A reviewer-paper assignment is then sampled using a randomized procedure that iteratively redistributes the probability mass placed on each reviewer-paper pair until all probabilities are either zero or one.

**Review Quality.** The above assignments are chosen based on maximizing the (expected) similarities of assigned reviewer-paper pairs, but those similarities may not be accurate proxies for the quality of review that the reviewer can provide for that paper. In practice, automated similarity-based assignments result in numerous complaints of low-expertise paper assignments from both authors and reviewers [3]. Meanwhile, self-reported assessments of reviewer-paper assignment quality can be collected from the reviewers themselves after the review. Conferences often ask reviewers to score their *expertise* in the paper's topic and/or *confidence* in their review [6, 29, 38]. Other indicators of review quality can also be considered; e.g., some conferences ask "meta-reviewers" or other reviewers

to evaluate the quality of written reviews directly [38, 39]. In this work, we consider self-reported expertise and confidence as our measures of review quality.

## 3 Off-Policy Evaluation

One attractive property of the randomized assignment described above is that while only one reviewer-paper assignment is sampled and deployed, many other assignments could have been sampled, and those assignments could equally well have been generated by some alternative assignment policy. The positive probability of other assignments allows us to investigate whether alternative assignment policies might have resulted in higher-quality reviews.

Let $A$ be a randomized assignment policy with a probability density $P_A$, where $\sum_{Z \in \{0,1\}^{|\mathcal{R}| \times |\mathcal{P}|}} P_A(Z) = 1$; $P_A(Z) \geq 0$, $\forall Z$; and $P_A(Z) > 0$ only for feasible assignments $Z$. Let $B$ be another policy with density $P_B$, defined similarly. We denote by $P_A(Z_i)$ and $P_B(Z_i)$ the marginal probabilities of assigning reviewer-paper pair $i$ under $A$ and $B$ respectively. Finally, let $Y_i \in \mathbb{R}$, where $i = (r, p) \in \mathcal{R} \times \mathcal{P}$, be the measure of the quality of reviewer $r$'s review of paper $p$.

We follow the potential outcomes framework for causal inference [40]. Throughout this work, we will let $A$ be the on-policy or the logging policy, i.e., the policy that the review data was collected under, while $B$ will denote one of several alternative policies of interest. In Section 5, we will describe the specific alternative policies we consider in this work. Define $N = \sum_{i \in \mathcal{R} \times \mathcal{P}} Z_i$ as the total number of reviews, fixed across policies and set ahead of time. We are interested in the following estimands:

$$
\mu_A = \mathbb{E}_{Z \sim P_A}\left[\frac{1}{N} \sum_{i \in \mathcal{R} \times \mathcal{P}} Y_i Z_i\right], \quad \mu_B = \mathbb{E}_{Z \sim P_B}\left[\frac{1}{N} \sum_{i \in \mathcal{R} \times \mathcal{P}} Y_i Z_i\right],
$$

where $\mu_A$ and $\mu_B$ are the expected review quality under policy $A$ and $B$, respectively.

In practice, we do not have access to all $Y_i$, but only those that were assigned. Let $Z^A \in \{0,1\}^{|\mathcal{R}| \times |\mathcal{P}|}$ be the assignment sampled under the on-policy A, drawn from $P_A$. We define the following Horvitz-Thompson estimators of the means:

$$
\widehat{\mu}_A = \frac{1}{N} \sum_{i \in \mathcal{R} \times \mathcal{P}} Y_i Z_i^A, \quad \widehat{\mu}_B = \frac{1}{N} \sum_{i \in \mathcal{R} \times \mathcal{P}} Y_i Z_i^A W_i, \quad \text{where } W_i = \frac{P_B(Z_i)}{P_A(Z_i)} \ \forall i \in \mathcal{R} \times \mathcal{P}. \quad (1)
$$

For now, suppose that $B$ has positive probability only where $A$ is positive (also known as satisfying "positivity"): $P_A(Z_i) > 0$ for all $i \in \mathcal{R} \times \mathcal{P}$ where $P_B(Z_i) > 0$. Then, all weights $W_i$ where $P_B(Z_i) > 0$ are bounded. As we will see, many policies of interest $B$ go beyond the support of $A$.

Under the positivity assumption, $\widehat{\mu}_A$ and $\widehat{\mu}_B$ are unbiased estimators of $\mu_A$ and $\mu_B$ respectively [41]. Moreover, the Horvitz-Thompson estimator is admissible in the class of all unbiased estimators [42]. Note that $\widehat{\mu}_A$ is the empirical mean of the observed assignment sampled on-policy, and $\widehat{\mu}_B$ is a weighted mean of the observed assignment based on inverse probability weighting. These estimators also rely on a standard causal inference assumption of "no interference"; i.e., that the outcomes $Y_i$ do not depend on the assignments $Z_j^A$ for any other reviewer-paper pair $j \neq i$. In Appendix B, we discuss the implications of this assumption in the peer review context.

**Challenges.** In off-policy evaluation, we are interested in evaluating a policy $B$ based on data collected under policy $A$. However, our ability to do so is typically limited to policies where the assignments that would be made under $B$ are possible under $A$. In practice, many interesting policies step outside of the support of $A$. Outcomes for reviewer-paper pairs outside the support of $A$ but with positive probability under $B$ ("positivity violations") cannot be estimated and must either be imputed by some model or have their contribution to the average outcome ($\mu_B$) bounded.

In addition to positivity violations, we identify three other mechanisms through which missing data with potential confounding may arise in the peer review context: absent reviewers, selective attrition, and manual reassignments. For absent reviewers, i.e., reviewers who have not submitted *any* reviews, we do not have a reason to believe that the reviews are missing due to the quality of the reviewer-paper assignment. Hence, we assume that their reviews are missing at random, and impute them with the weighted mean outcome of the observed reviews. For selective attrition, i.e., when some but not all reviews are completed, we instead employ conservative bounding techniques as for policy-based positivity violations. Finally, reviews might be missing due to manual reassignments by the program

chairs, after the assignment has been sampled. As a result, the originally assigned reviews will be missing and new reviews will be added. In such cases, we treat removed assignments as attrition (i.e., bounding their contribution) and ignore the newly introduced assignments as they did not arise from any determinable process. Concretely, we partition the reviewer-paper pairs into the following (mutually exclusive and exhaustive) sets:

▷ $\mathcal{I}^-$: positivity violations, $\{i = (r, p) \in \mathcal{R} \times \mathcal{P} : P_A(Z_i) = 0 \wedge P_B(Z_i) > 0\}$,

▷ $\mathcal{I}^{Abs}$: missing reviews where the reviewer was absent (submitted no reviews),

▷ $\mathcal{I}^{Att}$: remaining missing reviews, and

▷ $\mathcal{I}^+$: remaining pairs without positivity violations or missing reviews, $(\mathcal{R} \times \mathcal{P}) \backslash (\mathcal{I}^{Att} \cup \mathcal{I}^{Abs} \cup \mathcal{I}^-)$.

In the next section, we present methods for imputing or bounding the contribution of $\mathcal{I}^-$ to the estimate of $\widehat{\mu}_B$, and $\mathcal{I}^{Abs}$ and $\mathcal{I}^{Att}$ to the estimates of $\widehat{\mu}_A$ and $\widehat{\mu}_B$.

## 4 Imputation and Partial Identification

In the previous section, we defined three sets of reviewer-paper pairs $i$ for which outcomes $Y_i$ must be imputed rather than estimated: $\mathcal{I}^-, \mathcal{I}^{Abs}, \mathcal{I}^{Att}$. In this section, we describe varied methods for imputing these outcomes that rely on different strengths of assumptions, including methods that output point estimates and methods that output lower and upper bounds of $\widehat{\mu}_B$. In Section 5, we apply these methods to peer-review data from two computer science venues.

For missing reviews where the reviewer is absent ($\mathcal{I}^{Abs}$), we assume that the reviewer did not participate in the review process for reasons unrelated to the assignment quality (e.g., too busy). Specifically, we assume that the reviewers are missing at random and thus impute the mean outcome among $\mathcal{I}^+$, the pairs with no positivity violations or missing reviews: $\overline{Y} = (\sum_{i \in \mathcal{I}^+} Y_i Z_i^A W_i)/(\sum_{i \in \mathcal{I}^+} Z_i^A W_i)$.

In contrast, for positivity violations ($\mathcal{I}^-$) and the remaining missing reviews ($\mathcal{I}^{Att}$), we allow for the possibility that these reviewer-paper pairs being unobserved is correlated with their unobserved outcome. Thus, we consider imputing arbitrary values for $i$ in these subsets, which we denote by $Y_i^{Impute}$ and place into a matrix $Y^{Impute} \in \mathbb{R}^{|\mathcal{R}| \times |\mathcal{P}|}$, leaving entries for $i \notin \mathcal{I}^- \cup \mathcal{I}^{Att}$ undefined. This strategy corresponds to setting $Y_i = Y_i^{Impute}$ for $i \in \mathcal{I}^- \cup \mathcal{I}^{Att}$ in estimator (1). To obtain bounds, we impute both the assumed minimal and maximal values of $Y_i^{Impute}$.

These modifications result in a Horvitz-Thompson off-policy estimator with imputation. To denote this, we redefine $\widehat{\mu}_B$ to be a function $\widehat{\mu}_B : \mathbb{R}^{|\mathcal{R}| \times |\mathcal{P}|} \to \mathbb{R}$, where $\widehat{\mu}_B(Y^{Impute})$ denotes the estimator resulting from imputing entries from a particular choice of $Y^{Impute}$:

$$\widehat{\mu}_B(Y^{Impute}) = \frac{1}{N} \left( \sum_{i \in \mathcal{I}^+} Y_i Z_i^A W_i + \sum_{i \in \mathcal{I}^{Att}} Y_i^{Impute} Z_i^A W_i + \sum_{i \in \mathcal{I}^{Abs}} \overline{Y} Z_i^A W_i + \sum_{i \in \mathcal{I}^-} Y_i^{Impute} P_B(Z_i) \right).$$

The estimator computes the weighted mean of the observed ($Y_i$) and imputed outcomes ($Y_i^{Impute}$ and $\overline{Y}$). We impute $Y_i^{Impute}$ for the attrition ($\mathcal{I}^{Att}$) and positivity violation ($\mathcal{I}^-$) pairs, and $\overline{Y}$ for the absent reviewers ($\mathcal{I}^{Abs}$). Note that we weight the imputed positivity violations ($\mathcal{I}^-$) by $P_B(Z_i)$ rather than $Z_i W_i$, since the latter is undefined. Under the assumption that the imputed outcomes are accurate, $\widehat{\mu}_B(Y^{Impute})$ is an unbiased estimator of $\mu_B$. We can further construct confidence intervals using the methods described in Appendix C.

Next, we detail several methods by which we choose $Y^{Impute}$. These methods rely on various assumptions of different strength about the unobserved outcomes.

**Mean Imputation.** As a first approach, we assume that the mean outcome within $\mathcal{I}^+$ is representative of the mean outcome among the other pairs. This is a strong assumption, since the presence of a pair in $\mathcal{I}^-$ or $\mathcal{I}^{Att}$ may not be independent of their outcome. For example, if reviewers choose not to submit reviews when the assignment quality is poor, $\overline{Y}$ is not representative of the outcomes in $\mathcal{I}^{Att}$. Nonetheless, under this strong assumption, we can simply impute the mean outcome $\overline{Y}$ for all pairs necessitating imputation. Setting $Y_i^{Impute} = \overline{Y}$ for all $i \in \mathcal{I}^- \cup \mathcal{I}^{Att}$, we consider the following point estimate of $\mu_B$: $\widehat{\mu}_B(\overline{Y})$. While following from an overly strong assumption, we find it useful to compare our findings under this assumption to findings under subsequent weaker assumptions.

**Model Imputation.** Instead of simply imputing the mean outcome, we can assume that the unobserved outcomes $Y_i$ are some simple function of known covariates $X_i$ for each reviewer-paper

pair $i$. If so, we can directly estimate this function using a variety of statistical models, resulting in a point estimate of $\mu_B$. In doing so, we implicitly take on the assumptions made by each model, which determine how to generalize the covariate-outcome mapping from the observed pairs to the unobserved pairs. These assumptions are typically quite strong, since this mapping may be very different between the observed pairs (typically good matches) and unobserved pairs (typically less good matches).

Specifically, given the set of all observed reviewer-paper pairs $\mathcal{O} = \{i \in \mathcal{I}^+ : Z_i^A = 1\}$, we train a model $m$ using the observed data $\{(X_i, Y_i) : i \in \mathcal{O}\}$. Let $\widehat{Y}^{(m)} \in \mathbb{R}^{|\mathcal{R}| \times |\mathcal{P}|}$ denote the outcomes predicted by that model for each pair. We then consider $\widehat{\mu}_B(\widehat{Y}^{(m)})$ as a point estimate of $\mu_B$. In our experiments, we employ standard methods for classification, ordinal regression, and collaborative filtering: logistic regression (*clf-logistic*); ridge classification (*clf-ridge*); ordered logit (*ord-logit*); ordered probit (*ord-probit*); SVD++ collaborative filtering (*cf-svd++*); and K-nearest-neighbors collaborative filtering (*cf-knn*). Note that, unlike the other methods, the methods based on collaborative filtering model the missing data by using only the observed reviewer-paper outcomes ($Y$). We discuss our choice of methods, hyperparameters, and implementation details in Appendix E.

**Manski Bounds.** As a more conservative approach, we can exploit the fact that the outcomes $Y_i$ are bounded, letting us bound the mean of the counterfactual policy without making any assumptions on how the positivity violations arise. Such bounds are often called *Manski bounds* [17] in the econometrics literature on partial identification. To employ Manski bounds, we assume that all outcomes $Y$ can take only values between $y_{\min}$ and $y_{\max}$, e.g., self-reported expertise and confidence scores are limited to a pre-specified range on the review questionnaire. Then, setting $Y_i^{Impute} = y_{\min}$ or $Y_i^{Impute} = y_{\max}$ for all $i \in \mathcal{I}^- \cup \mathcal{I}^{Att}$, we can estimate the upper and lower bound of $\mu_B$ as $\widehat{\mu}_B(y_{\min})$ and $\widehat{\mu}_B(y_{\max})$. In Appendix C, we discuss how we construct confidence intervals for these estimates that asymptotically contain the true value of $\mu_B$ with probability at least $95\%$ by adapting the inference procedure proposed in [43].

**Monotonicity and Lipschitz Smoothness.** We now propose two styles of weak assumptions on the covariate-outcome mapping that can be leveraged to achieve tighter bounds on $\widehat{\mu}_B$ than the Manski bounds. In contrast to the strong modeling assumptions used in the mean and model imputation, these assumptions can be more intuitively understood and justified as conservative assumptions given particular choices of covariates.

*Monotonicity.* The first weak assumption we consider is a *monotonicity* condition. Intuitively, monotonicity captures the idea that we expect higher expertise for a reviewer-paper pair when some covariates are higher, all else equal. For example, in our experiments, we use the similarity component scores (bids, text similarity, subject area match) as covariates. Specifically, for covariate vectors $X_i$ and $X_j$, define the *dominance* relationship $X_i \succ X_j$ to mean that $X_i$ is greater than or equal to $X_j$ in all components and $X_i$ is strictly greater than $X_j$ in at least one component. Then, the monotonicity assumption states that: if $X_i \succ X_j$, then $Y_i \geq Y_j, \forall (i, j) \in (\mathcal{R} \times \mathcal{P})^2$.

Using this assumption to restrict the range of possible values for the unobserved outcomes, we seek upper and lower bounds on $\mu_B$. Recall that $\mathcal{O}$ is the set of all observed reviewer-paper pairs. One challenge is that the observed outcomes themselves ($Y_i$ for $i \in \mathcal{O}$) may violate the monotonicity condition. Thus, to find an upper or lower bound, we compute *surrogate values* $\widetilde{Y}_i \in \mathbb{R}$ that satisfy the monotonicity constraint for all $i \in \mathcal{O} \cup \mathcal{I}^{Att} \cup \mathcal{I}^{Abs} \cup \mathcal{I}^-$ while ensuring that the surrogate values $\widetilde{Y}_i$ for $i \in \mathcal{O}$ are as close as possible to the outcomes $Y_i$. The surrogate values $\widetilde{Y}_i$ for $i \in \mathcal{I}^{Att} \cup \mathcal{I}^-$ can then be imputed as outcomes.

Inspired by isotonic regression [44], we implement a two-level optimization problem. The primary objective minimizes the $\ell_1$ distance between $\widetilde{Y}_i$ and $Y_i$ for pairs with observed outcomes $i \in \mathcal{O}$. The second objective either minimizes (for a lower bound) or maximizes (for an upper bound) the sum of $Y_i$ for the unobserved pairs $i \in \mathcal{I}^{Att} \cup \mathcal{I}^{Abs} \cup \mathcal{I}^-$, weighted as in $\widehat{\mu}_B$. Define the universe of relevant pairs $\mathcal{U} = \mathcal{O} \cup \mathcal{I}^{Att} \cup \mathcal{I}^{Abs} \cup \mathcal{I}^-$ and define $\Psi$ as a very large constant. This results in the following

pair of optimization problems:

$$(\widetilde{Y}_{min}^{M}, \widetilde{Y}_{max}^{M}) = \underset{\widetilde{Y}_i : i \in \mathcal{U}}{\arg\min} \quad \Psi \sum_{i \in \mathcal{O}} |\widetilde{Y}_i - Y_i| \pm \left( \sum_{i \in \mathcal{I}^{Att} \cup \mathcal{I}^{Abs}} \widetilde{Y}_i W_i + \sum_{i \in \mathcal{I}^-} \widetilde{Y}_i P_B(Z_i) \right), \quad (2)$$

$$\text{s.t.} \quad \widetilde{Y}_i \geq \widetilde{Y}_j \quad \forall (i,j) \in \{\mathcal{U}^2 : X_i \succ X_j\}, \qquad y_{\min} \leq \widetilde{Y}_i \leq y_{\max} \quad \forall i \in \mathcal{U}.$$

The sign of the second objective term depends on whether a lower (negative) or upper (positive) bound is being computed. The last set of constraints corresponds to the same constraints used to construct the Manski bounds described earlier, which is combined here with monotonicity to jointly constrain the possible outcomes. The above problem can be reformulated and solved as a linear program using standard techniques. We can then estimate the upper and lower bound of $\mu_B$ as $\widehat{\mu}_B(\widetilde{Y}_{min}^{Mon})$ and $\widehat{\mu}_B(\widetilde{Y}_{max}^{Mon})$, and construct 95% confidence intervals as discussed in Appendix C.

*Lipschitz Smoothness.* The second weak assumption we consider is a *Lipschitz smoothness* assumption on the correspondence between covariates and outcomes. Intuitively, this captures the idea that we expect two reviewer-paper pairs who are very similar in covariate space to have similar expertise. For covariate vectors $X_i$ and $X_j$, define $d(X_i, X_j)$ as some notion of distance between the covariates. Then, the Lipschitz assumption states that there exists a constant $L$ such that $|Y_i - Y_j| \leq L d(X_i, X_j)$ for all $(i, j) \in (\mathcal{R} \times \mathcal{P})^2$. In practice, we can choose an appropriate value of $L$ by studying the many pairs of observed outcomes in the data (Section 5 and Appendix G), though this approach assumes that the Lipschitz smoothness of the covariate-outcome function is the same for observed and unobserved pairs.

As in the previous section, we introduce surrogate values $\widetilde{Y}_i \in \mathbb{R}$ and implement a two-level optimization problem to address Lipschitz violations within the observed outcomes (i.e., if two observed pairs are very close in covariate space but have different outcomes). The resulting pair of optimization problems is defined exactly as in problem (2), except that the first set of constraints is replaced with constraints $|\widetilde{Y}_i - \widetilde{Y}_j| \leq L d(X_i, X_j), \forall (i, j) \in \mathcal{U}$. We include the complete LP formulation in Appendix D. We denote the solutions to these problems by $\widetilde{Y}_{min}^{L}$ and $\widetilde{Y}_{max}^{L}$. As before, the sign of the second objective term depends on whether a lower (negative) or upper (positive) bound is being computed. The last set of constraints are again the same constraints used to construct the Manski bounds described earlier, which here are combined with the Lipschitz assumption to jointly constrain the possible outcomes. In the limit, as $L \to \infty$, the Lipschitz constraints become vacuous and we recover the Manski bounds. This problem can again be reformulated and solved as a linear program using standard techniques. We can then estimate the upper and lower bound of $\mu_B$ as $\widehat{\mu}_B(\widetilde{Y}_{min}^{L})$ and $\widehat{\mu}_B(\widetilde{Y}_{max}^{L})$, and construct 95% confidence intervals as discussed in Appendix C.

## 5 Experiments

We apply our framework to data from two venues that used randomized paper assignments as described in Section 2, the 2021 Workshop on Theory and Practice of Differential Privacy (*TPDP*) and the 2022 AAAI Conference on Advancement in Artificial Intelligence (*AAAI*). In both settings, we aim to understand the effect that changing parameters of the assignment policies would have on review quality. The analyses were approved by our Institutional Review Boards.

**Datasets.**
*TPDP.* The TPDP workshop received 95 submissions and had a pool of 35 reviewers. Each paper received exactly 3 reviews, and reviewers were assigned 8 or 9 reviews, for a total of 285 assigned reviewer-paper pairs. The reviewers were asked to bid on the papers and could place one of the following bids (the corresponding value of $B_i$ is shown in the parenthesis): *"very low"* $(-1)$, *"low"* $(-0.5)$, *"neutral"* $(0)$, *"high"* $(0.5)$, or *"very high"* $(1)$, with *"neutral"* as the default. The similarity for each reviewer-paper pair was defined as a weighted sum of the bid score, $B_i$, and text-similarity scores, $T_i$: $S_i = w_{\text{text}} T_i + (1 - w_{\text{text}}) B_i$, with $w_{\text{text}} = 0.5$. The randomized assignment was run with an upper bound of $q = 0.5$. In their review, the reviewers were asked to assess the alignment between the paper and their *expertise* (between 1: irrelevant and 4: very relevant), and to report their review *confidence* (between 1: educated guess and 5: absolutely certain). We consider these two responses as our measures of quality.

*AAAI.* In the AAAI conference, submissions were assigned to reviewers in multiple sequential stages across two rounds of submissions. We examine the stage of the first round where the randomized assignment algorithm was used to assign all submissions to a pool of "senior reviewers." The assignment involved 8,450 papers and 3,145 reviewers; each paper was assigned to one reviewer, and each reviewer was assigned at most 3 or 4 papers based on their primary subject area. The similarity $S_i$ for every reviewer-paper pair $i$ was based on three scores: text-similarity $T_i \in [0, 1]$, subject-area score $K_i \in [0, 1]$, and bid $B_i$. Bids were chosen from the following list (with the corresponding value of $B_i$ shown in the parenthesis, where $\lambda_{\text{bid}} = 1$ is a parameter scaling the impact of positive bids as compared to neutral/negative bids): *"not willing"* (0.05), *"not entered"* (1), *"in a pinch"* $(1+0.5\lambda_{\text{bid}})$, *"willing"* $(1 + 1.5\lambda_{\text{bid}})$, *"eager"* $(1 + 3\lambda_{\text{bid}})$. The default option was *"not entered"*. Similarities were computed as: $S_i = (w_{\text{text}}T_i + (1 - w_{\text{text}})K_i)^{1/B_i}$, with $w_{\text{text}} = 0.75$. The actual similarities differed from this base similarity formula in a few special cases (e.g., missing data); we provide the full description of the similarity computation in Appendix F. The randomized assignment was run with $q = 0.52$. Reviewers reported an *expertise* score (between 0: not knowledgeable and 5: expert) and a *confidence* score (between 0: not confident and 4: very confident), which we consider as our quality measures.

**Assumption Suitability.** For both the monotonicity and Lipschitz assumptions (as well as the model imputations), we work with the covariates $X_i$, a vector of the two (TPDP) or three (AAAI) component scores used in the computation of similarities. In Appendix G, we consider whether these assumptions are reasonable with respect to our choice of covariates by evaluating them on the observed outcomes. For the monotonicity assumption, we find that it holds on large majorities of observed outcome pairs. For the Lipschitz assumption, we use a normalized $\ell_1$ distance and choose values of $L$ corresponding to less than 10%, 5%, and 1% violations on the observed outcomes.

**Results.** We now present the analyses of the two datasets using the methods introduced in Section 4. For brevity, we report our analysis using self-reported expertise as the quality measure $Y$, but include the results using self-reported confidence in Appendix L. When solving the LPs that output alternative randomized assignments (Appendix A), we often encounter multiple unique optimal solutions and employ a persistent arbitrary tie-breaking procedure to choose amongst them (Appendix H).

*TPDP.* We perform two analyses on the TPDP data, shown in Figure 1 (left). First, we analyze the choice of how to interpolate between the bids and the text similarity when computing the composite similarity score for each reviewer-paper pair. We examine a range of assignments, from an assignment based only on the bids ($w_{\text{text}} = 0$) to an assignment based only on the text similarity ($w_{\text{text}} = 1$), focusing our off-policy evaluation on deterministic assignments (i.e., policies with $q = 1$). Setting $w_{\text{text}} \in [0, 0.75]$ results in very similar assignments, each of which has Manski bounds overlapping with the on-policy (Figure 1A, left). Within this region, the models (Figure 1B, left), monotonicity bounds (Figure 1C, left), and Lipschitz bounds (Figure 1D, left) all agree that the expertise is similar to the on-policy. However, setting $w_{\text{text}} \in (0.75, 0.9)$ results in a significant improvement in average expertise, even without any additional assumptions (Figure 1A, left). Finally, setting $w_{\text{text}} \in (0.9, 1]$ leads to assignments that are significantly different from the assignments supported by the on-policy, which results in many positivity violations and wider confidence intervals, even under the monotonicity and Lipschitz smoothness assumptions (Figures 1A-D, left). Note that within this region, the models significantly disagree on the expertise (Figure 1B, left), indicating that the strong assumptions made by such models may not be accurate. Altogether, these results suggest that putting more weight on the text similarity (versus bids) leads to higher-expertise reviews.

Second, we investigate the "cost of randomization" to prevent fraud, measuring the effect of increasing $q$ and thereby reducing randomness in the optimized random assignment. We consider values between $q = 0.4$ and $q = 1$ (optimal deterministic assignment). Recall the on-policy has $q = 0.5$. When varying $q$, we find that except for a small increase in the region around $q = 0.75$, the average expertise for policies with $q > 0.5$ is very similar to that of the on-policy (Figures 1A-D, right). These results suggest that using a randomized instead of a deterministic policy does not lead to a significant reduction in self-reported expertise, an observation that should be contrasted with the previously documented reduction in the expected sum-similarity objective under randomized assignment [16]; see further analysis in Appendix I.

*AAAI.* We perform three analyses on the AAAI data, shown in Figure 1 (right). First, we examine the effect of interpolating between the text-similarity scores and the subject area scores by varying $w_{\text{text}} \in [0, 1]$, again considering only deterministic policies (i.e., $q = 1$). The on-policy sets

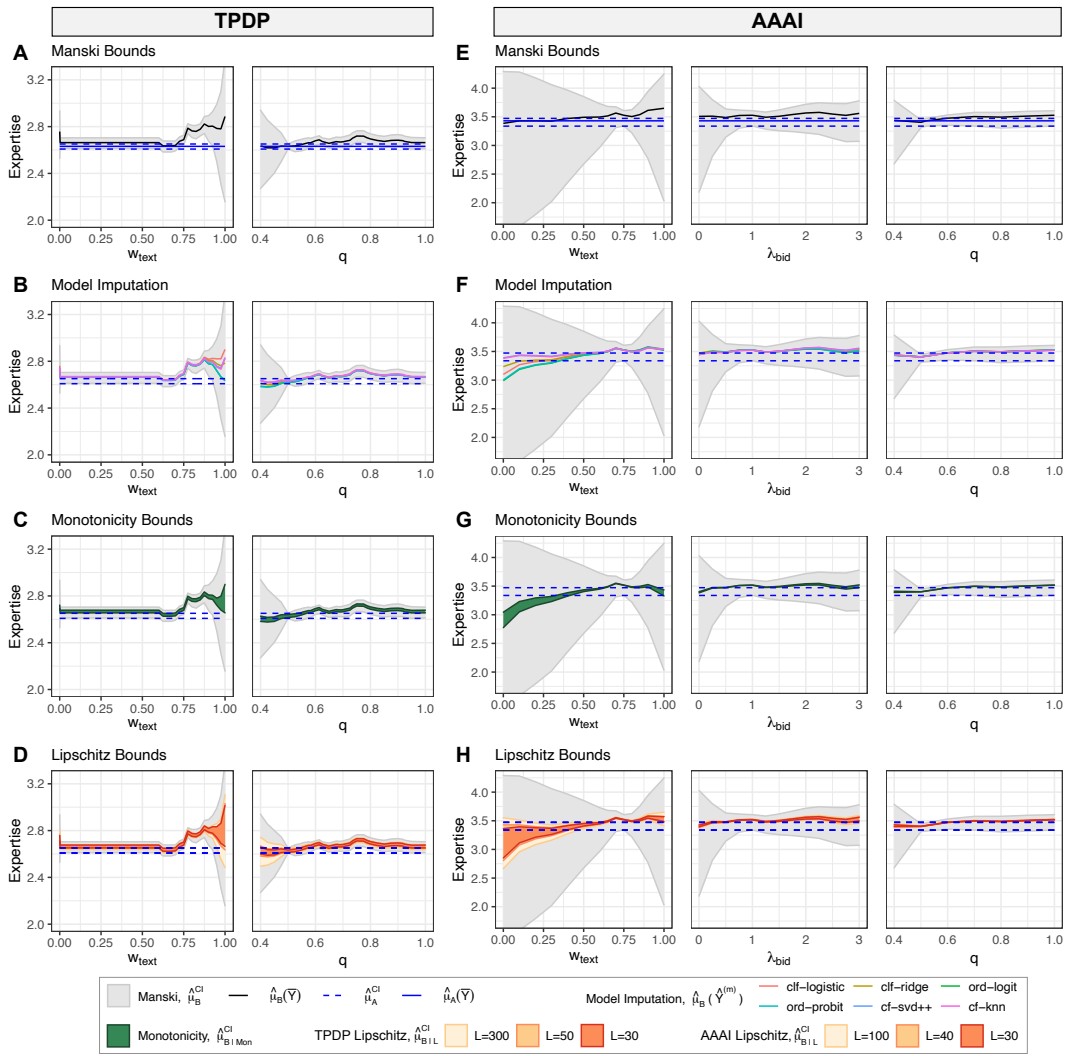

Figure 1: Expertise of off-policies varying $w_{\text{text}}$ and $q$ for TPDP and $w_{\text{text}}$, $\lambda_{\text{bid}}$, and $q$ for AAAI, computed using the different estimation methods described in Section 4. The dashed blue lines indicate Manski bounds around the on-policy expertise, and the grey lines indicate Manski bounds around the off-policy expertise. The error bands (denoted $\widehat{\mu}_B^{CI}$ for the Manski bounds, $\widehat{\mu}_{B|M}^{CI}$ for the monotonicity bounds, and $\widehat{\mu}_{B|L}^{CI}$ for the Lipschitz bounds) represent confidence intervals that asymptotically contain the true value of $\mu_B$ with probability at least 95%, as described in Appendix C. We observe similar patterns using confidence as an outcome, as shown in Figure 7 in the Appendix. Note that to focus on the most relevant regions of the plots, the vertical axes do not start at zero.

$w_{\text{text}} = 0.75$. Due to large numbers of positivity violations, the Manski bounds are uninformative (Figure 1E, left), so we turn to the other estimators. The model imputation analysis indicates that policies with $w_{\text{text}} \geq 0.75$ may have slightly higher expertise than the on-policy and indicates lower expertise in the region where $w_{\text{text}} \leq 0.5$ (Figure 1F, left). However, the models differ somewhat in their predictions for low $w_{\text{text}}$, indicating that the assumptions made by these models may not be reliable. The monotonicity bounds more clearly indicate low expertise compared to the on-policy when $w_{\text{text}} \leq 0.25$, but are also slightly more pessimistic about the $w_{\text{text}} \geq 0.75$ region than the models (Figure 1G, left). The Lipschitz bounds indicate slightly higher than on-policy expertise for $w_{\text{text}} \geq 0.75$ and potentially suggest slightly lower than on-policy expertise for $w_{\text{text}} \leq 0.25$ (Figure 1H, left). Overall, all methods of analysis indicate that low values of $w_{\text{text}}$ result in worse assignments, but the effect of considerably increasing $w_{\text{text}}$ is unclear.

Second, we examine the effect of increasing the weight on positive bids by varying the values of $\lambda_{\text{bid}}$. Recall that $\lambda_{\text{bid}} = 1$ corresponds to the on-policy and a higher (respectively lower) value of $\lambda_{\text{bid}}$ indicates greater (respectively lesser) priority given to positive bids relative to neutral/negative bids. We investigate policies that vary $\lambda_{\text{bid}}$ within the range $[0, 3]$, and again consider only deterministic policies (i.e., $q = 1$). The Manski bounds are again too wide to be informative (Figure 1E, middle). The models all indicate similar values of expertise for all values of $\lambda_{\text{bid}}$ and are all slightly more optimistic about expertise than the Manski bounds around the on-policy (Figure 1F, middle). The monotonicity and Lipschitz bounds both agree that the $\lambda_{\text{bid}} \geq 1$ region has slightly higher expertise as compared to the on-policy (Figure 1G-H, middle). Overall, our analyses provide some indication that increasing $\lambda_{\text{bid}}$ may result in slightly higher levels of expertise.

Finally, we also examine the effect of varying $q$ within the range $[0.4, 1]$. Recall that the on-policy sets $q = 0.52$. We see that the models (Figure 1F, right), the monotonicity bounds (Figure 1G, right), and the Lipschitz bounds (Figure 1H, right) all strongly agree that the region $q \geq 0.6$ has slightly higher expertise than the region $q \in [0.4, 0.6]$. However, the magnitude of this change is small, indicating that the "cost of randomization" is not very significant.

**Power Investigation: Purposefully Bad Policies.** As many of the off-policy assignments we consider have relatively similar estimated quality, we also ran additional analyses to show that our methods can discern differences between good policies (optimized toward high reviewer-paper similarity assignments) and policies intentionally chosen to have poor quality ("optimized" toward low reviewer-paper similarity assignments). We refer the reader to Appendix J for further discussion.

## 6   Discussion and Conclusion

In this work, we evaluate the quality of off-policy reviewer-paper assignments in peer review using data from two venues that deployed randomized reviewer assignments. We propose new techniques for partial identification that allow us to draw useful conclusions about the off-policy review quality, even in the presence of large numbers of positivity violations and missing reviews. For a more extensive treatment of the methods proposed in this work, we refer the reader to Khan et al. [45].

**Limitations.** One limitation of off-policy evaluation is that our ability to make inferences inherently depends on the amount of randomness introduced on-policy. For instance, if there is a small amount of randomness, we will be able to estimate only policies that are relatively close to the on-policy unless we are willing to make some assumptions. The approaches presented in this work allow us to examine the strength of the evidence under a wide range of types and strengths of assumptions (model imputation, boundedness of the outcome, monotonicity, and Lipschitz smoothness) and to test whether these assumptions lead to converging conclusions. We emphasize that these assumptions (except for the boundedness of the outcome) are fundamentally unverifiable in our setting, given that the outcomes for unassigned pairs are unobserved. Thus, our resulting conclusions about the quality of alternative assignments cannot be directly verified without actually deploying the off-policies in question. Additionally, the measurement of review quality is a difficult question for any evaluation method: our analyses use the reviewers' self-reported expertise and confidence scores as proxies for review quality, but the self-reported nature of these measures makes them subject to various biases. Our conclusions should be interpreted in light of the possibility of misalignment between self-reported expertise/confidence and the true review quality. Finally, our substantive conclusions are based on analyses of data from two venues, and thus further work is needed to test their generalizability.

**Future Work.** In the context of peer review, the present work considers only a few parameterized slices of the vast space of reviewer-paper assignment policies, while there are many other substantive questions that our methodology can be used to answer. For instance, one could evaluate assignment quality under a different method of computing similarity scores (e.g., different NLP algorithms [46]), additional constraints on the assignment (e.g., based on seniority or geographic diversity [6]), or objective functions other than the sum-of-similarities (e.g., various fairness-based objectives [35, 36, 47, 48]). Additional thought should also be given to the trade-offs between maximizing review quality and broader considerations of reviewer welfare: while assignments based on high text similarity may yield slightly higher-quality reviews, reviewers may be more willing to review again if the assignment policy follows their bids more closely. Beyond peer review, our work is applicable to off-policy evaluation in other matching problems, including education [49, 50], advertising [21], and ride-sharing [22].

## Acknowledgments and Disclosure of Funding

We thank Gautam Kamath and Rachel Cummings for allowing us to conduct this study in TPDP and Melisa Bok and Celeste Martinez Gomez from OpenReview for helping us with the OpenReview APIs. We are also grateful to Samir Khan and Tal Wagner for helpful discussions. This work was supported in part by NSF CAREER Award 2143176, NSF CAREER Award 1942124, NSF CIF 1763734, and ONR N000142212181.

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

# Counterfactual Evaluation of Peer-Review Assignment Policies
## Supplemental Material

**Martin Saveski, Steven Jecmen, Nihar B. Shah, Johan Ugander**

## A  Linear Programs for Peer-Review Assignment

**Deterministic Assignment.**  Let $Z \in \{0,1\}^{|\mathcal{R}| \times |\mathcal{P}|}$ be an assignment matrix where $Z_{r,p}$ denotes whether reviewer $r \in \mathcal{R}$ is assigned to paper $p \in \mathcal{P}$. Given a matrix of similarity scores $S \in \mathbb{R}_{\geq 0}^{|\mathcal{R}| \times |\mathcal{P}|}$, a standard objective is to find an assignment of papers to reviewers that maximizes the sum of similarities of the assigned pairs, subject to constraints that each paper is assigned to an appropriate number of reviewers $\ell$, each reviewer is assigned no more than a maximum number of papers $k$, and conflicts of interest are respected [5, 27, 30–34]. Denoting the set of conflict-of-interest pairs by $\mathcal{C} \subset \mathcal{R} \times \mathcal{P}$, this optimization problem can be formulated as the following linear program:

$$\max_{Z_{r,p}:r\in\mathcal{R},p\in\mathcal{P}} \sum_{r\in\mathcal{R},p\in\mathcal{P}} Z_{r,p} S_{r,p}$$

$$\text{s.t.} \quad \sum_{r\in\mathcal{R}} Z_{r,p} = \ell \qquad \forall p \in \mathcal{P}$$

$$\sum_{p\in\mathcal{P}} Z_{r,p} \leq k \qquad \forall r \in \mathcal{R}$$

$$Z_{r,p} = 0 \qquad \forall (r,p) \in \mathcal{C}$$

$$0 \leq Z_{r,p} \leq 1 \qquad \forall r \in \mathcal{R}, p \in \mathcal{P}.$$

By total unimodularity conditions, this problem has an optimal solution where $Z_{r,p} \in \{0,1\}, \forall r \in \mathcal{R}, p \in \mathcal{P}$.

Although the above strategy is the primary method used for paper assignments in large-scale peer review, other variants of this method have been proposed and used in the literature. These algorithms consider various properties in addition to the total similarity, such as fairness [35, 36], strategyproofness [37, 51], envy-freeness [47] and diversity [52]. We focus on the sum-of-similarities objective here, but our off-policy evaluation framework is agnostic to the specific objective function.

**Randomized Assignment.**  As one approach to strategyproofness, Jecmen et al. [16] introduce the idea of using randomization to prevent colluding reviewers and authors from being able to guarantee their assignments. Specifically, the algorithm computes a randomized paper assignment, where the marginal probability $P(Z_{r,p})$ of assigning any reviewer $r$ to any paper $p$ is at most a parameter $q \in [0,1]$, chosen a priori by the program chairs. These marginal probabilities are determined by the following linear program, which maximizes the expected similarity of the assignment:

$$\max_{P(Z_{r,p}):r\in\mathcal{R},p\in\mathcal{P}} \sum_{r\in\mathcal{R},p\in\mathcal{P}} P(Z_{r,p}) S_{r,p} \qquad (3)$$

$$\text{s.t.} \quad \sum_{r\in\mathcal{R}} P(Z_{r,p}) = \ell \qquad \forall p \in \mathcal{P}$$

$$\sum_{p\in\mathcal{P}} P(Z_{r,p}) \leq k \qquad \forall r \in \mathcal{R}$$

$$P(Z_{r,p}) = 0 \qquad \forall (r,p) \in \mathcal{C}$$

$$0 \leq P(Z_{r,p}) \leq q \qquad \forall r \in \mathcal{R}, p \in \mathcal{P}.$$

A reviewer-paper assignment is then sampled using a randomized procedure that iteratively redistributes the probability mass placed on each reviewer-paper pair until all probabilities are either zero or one. This procedure ensures only that the desired marginal assignment probabilities are satisfied, providing no guarantees on the joint distributions of assigned pairs.

## B  "No Interference" Assumption

Our estimators assume that there is no interference between the units, i.e., that the treatment of one unit does not affect the outcomes for the other units. In the causal inference literature, this assumption

is referred to as the Stable Unit Treatment Value Assumption (SUTVA) [53]. In the context of peer review, SUTVA implies that: ($i$) The quality $Y_{r,p}$ of the review by reviewer $r$ reviewing paper $p$ does not depend on what other reviewers are assigned to paper $p$; and ($ii$) the quality also does not depend on the other papers that reviewer $r$ was assigned to review. The first assumption is quite realistic as in most peer review systems the reviewers cannot see other reviews until they submit their own. The second assumption is important to understand, as there could be "batch effects": a reviewer may feel more or less confident about their assessment (if measuring quality by confidence) depending on what other papers they were assigned to review. We do not test for batch effects or other violations of SUTVA in this work, which typically require either strong modeling assumptions or complex experimental designs [54–56] specifically tailored for testing SUTVA, but consider it important future work.

## C  Variance Computation and Confidence Interval Construction

In this section, we discuss how we construct confidence intervals around $\mu_B$ under Manski, monotonicity, and Lipschitz assumptions. To construct confidence intervals, we estimate the variance of $\widehat{\mu}_B(Y^{Impute})$ as follows:

$$\widehat{\text{Var}}[\widehat{\mu}_B(Y^{Impute})] = \frac{1}{N^2} \sum_{(i,j) \in (\mathcal{R} \times \mathcal{P})^2} \text{Cov}[Z_i, Z_j] Z_i^A Z_j^A W_i W_j Y_i' Y_j',$$

$$\text{where } Y_i' = \begin{cases} Y_i & \text{if } i \in \mathcal{I}^+ \\ Y_i^{Impute} & \text{if } i \in \mathcal{I}^{Att} \cup \mathcal{I}^- \\ \overline{Y} & \text{if } i \in \mathcal{I}^{Abs}. \end{cases}$$

The covariance terms (taken over $Z \sim P_A$) are not known exactly, owing to the fact that the procedure by Jecmen et al. [16] only constrains the marginal probabilities of individual reviewer-paper pairs, but pairs of pairs can be non-trivially correlated. In the absence of a closed-form expression, we use Monte Carlo methods to tightly estimate these covariances. In both our analyses of the TPDP and AAAI datasets, we sampled 1 million assignments and computed the empirical covariance. We ran an additional analysis to investigate the variability of our variance estimates. We took a bootstrap sample of 100,000 assignments (from the set of all 1 million assignments we sampled) and computed the variance based only on the (smaller) bootstrap sample. We repeated this procedure 1,000 times and computed the variance of our variance estimates. We found that the variance of our variance estimates is very small (less than $10^{-9}$) even when we use 10 times fewer sampled assignments, suggesting that we have sampled enough assignments to accurately estimate the variance.

When employing Manski bounds, we can adapt a well-established inference procedure to construct 95% confidence intervals of a partial identification region that asymptotically contain the true value of $\mu_B$ with probability at least 95%. Following Imbens and Manski [43], we construct the interval:

$$\widehat{\mu}_B^{CI} \in \left[ \widehat{\mu}_B(y_{\min}) - z_{\alpha,n}' \sqrt{\widehat{\text{Var}}[\widehat{\mu}_B(y_{\min})]/N}, \ \widehat{\mu}_B(y_{\max}) + z_{\alpha,n}' \sqrt{\widehat{\text{Var}}[\widehat{\mu}_B(y_{\max})]/N} \right],$$

where the $z$-score analog $z_{\alpha,n}'$ ($\alpha = 0.95$), is set by their procedure such that the interval asymptotically has at least 95% coverage under plausible regularity conditions; for further details, see the discussion below. When employing the monotonicity assumption, we construct the interval:

$$\widehat{\mu}_{B|M}^{CI} \in \left[ \widehat{\mu}_B(\widetilde{Y}_{min}^{Mon}) - z_{\alpha,n}' \sqrt{\widehat{\text{Var}}[\widehat{\mu}_B(\widetilde{Y}_{min}^{Mon})]/N}, \ \widehat{\mu}_B(\widetilde{Y}_{max}^{Mon}) + z_{\alpha,n}' \sqrt{\widehat{\text{Var}}[\widehat{\mu}_B(\widetilde{Y}_{max}^{Mon})]/N} \right],$$

and when employing the Lipschitz assumption, we construct the interval:

$$\widehat{\mu}_{B|L}^{CI} \in \left[ \widehat{\mu}_B(\widetilde{Y}_{min}^{L}) - z_{\alpha,n}' \sqrt{\widehat{\text{Var}}[\widehat{\mu}_B(\widetilde{Y}_{min}^{L})]/N}, \ \widehat{\mu}_B(\widetilde{Y}_{max}^{L}) + z_{\alpha,n}' \sqrt{\widehat{\text{Var}}[\widehat{\mu}_B(\widetilde{Y}_{max}^{L})]/N} \right].$$

These intervals converge uniformly to the specified $\alpha$-level coverage under a set of regularity assumptions on the behavior of the estimators of the upper and lower endpoints of the interval estimate: Assumption 1 from [43], establishing the coverage result in Lemma 4 there. It is difficult to verify whether Assumption 1 is satisfied for the designs (sampling reviewer-paper matchings) and interval endpoint estimators (Manski, monotonocity, Lipschitz) in this work.

A different set of assumptions, most significantly that the fraction of missing data is known before assignment, support a different method for computing confidence intervals with the coverage result in Lemma 3 from [43], obviating the need for Assumption 1. In our setting, small amounts of attrition (relative to the number of policy-induced positivity violations) mean that the fraction of data that is missing is not exactly known before assignment, but almost. In practice, we find that the Imbens-Manski interval estimates from their Lemma 3 (assuming a known fraction of missing data) and Lemma 4 (assuming Assumption 1) are nearly identical for all three of the Manksi-, monotonicity-, and Lipschitz-based estimates, suggesting the coverage is well-behaved. A detailed theoretical analysis of whether the estimators obey the regularity conditions of Assumption 1 is beyond the scope of this work.

# D  Linear Programs for Partial Identification Based on Lipschitz Smoothness

In this section, we provide a more detailed description of the linear programs used to compute our partial identification estimates assuming Lipschitz smoothness on the correspondence between covariates and outcomes. Intuitively, the Lipschitz smoothness assumption captures the idea that we expect two reviewer-paper pairs that are very similar in covariate space to have similar expertise. For covariate vectors $X_i$ and $X_j$, define $d(X_i, X_j)$ as some notion of distance between the covariates. Then, the Lipschitz assumption states that there exists a constant $L$ such that $|Y_i - Y_j| \leq Ld(X_i, X_j)$ for all $i$ and $j$ in $\mathcal{R} \times \mathcal{P}$.

As in our formulation for partial identification based on monotonicity (Section 4, LP (2)), we introduce surrogate values $\widetilde{Y}_i$ and implement a two-level optimization problem to address Lipschitz violations within the observed outcomes, i.e., if two observed pairs are very close in covariate space but have different outcomes. We also define the universe of relevant pairs $\mathcal{U} = \mathcal{O} \cup \mathcal{I}^{Att} \cup \mathcal{I}^{Abs} \cup \mathcal{I}^-$ and a very large constant $\Psi$.

This results in the following pair of optimization problems:

$$(\widetilde{Y}_{min}^L, \widetilde{Y}_{max}^L) = \underset{\widetilde{Y}_i : i \in \mathcal{U}}{\mathrm{argmin}} \, \Psi \sum_{i \in \mathcal{O}} |\widetilde{Y}_i - Y_i| \pm \left( \sum_{i \in \mathcal{I}^{Att} \cup \mathcal{I}^{Abs}} \widetilde{Y}_i W_i + \sum_{i \in \mathcal{I}^-} \widetilde{Y}_i P_B(Z_i) \right)$$
$$\text{s.t.} \quad |\widetilde{Y}_i - \widetilde{Y}_j| \leq Ld(X_i, X_j) \quad \forall (i, j) \in \mathcal{U},$$
$$y_{min} \leq \widetilde{Y}_i \leq y_{max} \qquad \forall i \in \mathcal{U}.$$

The sign of the second objective term depends on whether a lower (negative) or upper (positive) bound is being computed. The last set of constraints are the same constraints used to construct the Manski bounds described in Section 4, which here are combined with the Lipschitz assumption to jointly constrain the possible outcomes. In the limit, as $L \to \infty$, the Lipschitz constraints become vacuous and we recover the Manski bounds. This problem can be reformulated and solved as a linear program using standard techniques.

# E  Model Implementation

To impute the outcomes of the unobserved reviewer-paper pairs, we train classification, ordinal regression, and collaborative filtering models. Classification models are suitable since the reviewers select their expertise and confidence scores from a set of pre-specified choices. Ordinal regression models additionally model the fact that the scores have a natural ordering. Collaborative filtering models, in contrast to the classification and ordinal regression models, do not rely on covariates and instead model the structure of the observed entries in the reviewer-paper outcome matrix, which is akin to user-item rating matrices found in recommender systems.

In the classification and regression models, we use the covariates $X_i$ for each reviewer-paper pair as input features. In our analysis, we consider the two/three component scores used to compute the similarities: for TPDP, $X_i = (T_i, B_i)$; for AAAI, $X_i = (T_i, K_i, B_i)$. These are the primary components used by conference organizers to compute similarities, so we expect them to be usefully correlated with match quality. Although we perform our analysis with this choice of covariates, one

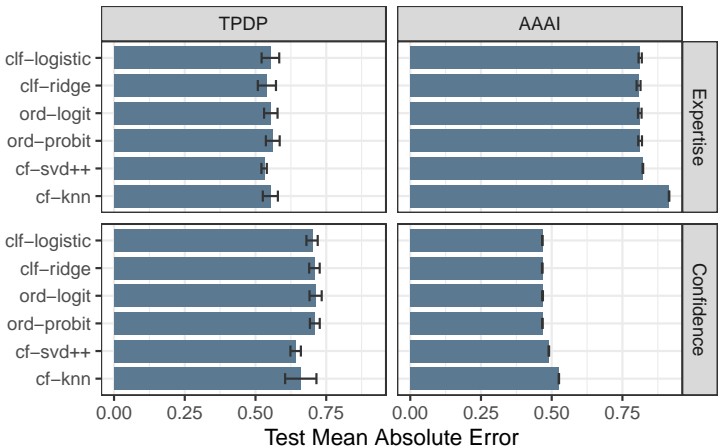

Figure 2: Test performance of the imputation models described in Section 4 (Model Imputation), averaged across 10 random train/test splits of all observed reviewer-paper pairs. The error bars show 95% confidence intervals.

could also include various other features of each reviewer-paper pair, e.g., some encoding of reviewer and paper subject areas, reviewer seniority, etc.

To evaluate the performance of the models, we randomly split the observed reviewer-paper pairs into train (75%) and test (25%) sets, fit the models on the train set, and measure the mean absolute error (MAE) of the predictions on the test set. To get more robust estimates of the performance, we repeat this process 10 times. In the training phase, we use 10-fold cross-validation to tune the hyperparameters, using MAE as a selection criterion, and retrain the model on the full training set with the best hyperparameters. We also consider two preprocessing decisions: (*a*) whether to encode the bids as one-hot categorical variables or continuous variables with the values described in Section 5, and (*b*) whether to standardize the features. In both cases, we used the settings that, overall, worked best (at prediction) for each model. We tested several models from each model category. To simplify the exposition, we only report the results of the two best-performing models in each category. The code repository referenced in Section 1 contains the implementation of all models, including the sets of hyperparameters considered for each model.

Figure 2 shows the test MAE across the 10 random train/test splits (means and 95% CIs) using expertise and confidence outcomes for both TPDP and AAAI. We note that all models perform significantly better than a baseline that predicts the mean outcome in the train set. For TPDP, we find that all models perform similarly, except for *cf-svd++*, which performs slightly better than the other models, both for expertise and confidence. For AAAI, all classification and regression models perform similarly, but the collaborative filtering models perform slightly worse. This difference in performance is perhaps due to the fact that we consider a larger set of covariates for AAAI than TPDP, likely making the classification and ordinal regression models more predictive.

Finally, to estimate $\widehat{\mu}_B$, we train each model on the set of all observed reviewer-paper pairs, predict the outcomes for all unobserved pairs, and impute the predicted outcomes as described in Section 4. In the training phase, we use 10-fold cross-validation to select the hyperparameters and refit the model on the full set of observed reviewer-paper pairs.

## F  Details of AAAI Assignment

In Section 5, we described a simplified version of the stage of the AAAI assignment procedure that we analyze, i.e., the assignment of senior reviewers to the first round of submissions. In this section, we describe this stage of the AAAI paper assignment more precisely.

A randomized assignment was computed between 3145 senior reviewers and 8450 first-round paper submissions, independent of all other stages of the reviewer assignment. The set of senior reviewers was determined based on reviewing experience and publication record; these criteria were external to the assignment. Each paper was assigned $\ell = 1$ senior reviewer. Reviewers were assigned to at

most $k = 4$ papers, with the exception of reviewers with a "Machine Learning" primary area or in the "AI For Social Impact" track, who were assigned to at most $k = 3$ papers. The probability limit was $q = 0.52$.

The similarities were computed from text-similarity scores $T_i$, subject-area scores $K_i$, and bids $B_i$. Either the text-similarity scores or the area scores could be missing for a given reviewer-paper pair, due to either a reviewer failing to provide the needed information or due to other errors in the computation of the scores. The text-similarity scores $T_i$ were created using text-based scores from two different sources: $(i)$ the Toronto Paper Matching System (TPMS) [27], and $(ii)$ the ACL Reviewer Matching code [28]. The text-similarity scores $T_i$ was set equal to the TPMS score for all pairs where this score was not missing, set equal to the ACL score for all other pairs where the ACL score was not missing, and marked as missing if both scores were missing. The subject-area scores were computed from reviewer and paper subject areas using the procedure described in Appendix A of [6].

Next, base scores $S_i' = w_{\text{text}}T_i + (1 - w_{\text{text}})K_i$ were then computed with $w_{\text{text}} = 0.75$, if both $T_i$ and $K_i$ were not missing. If either $T_i$ or $K_i$ was missing, the base score was equal to the non-missing score of the two. If both were missing, the base score was set as $S_i' = 0$. For pairs where the bid was *"willing"* or *"eager"* and $K_i = 0$, the base score was set as $S_i' = T_i$.

Next, final scores were computed as $S_i = S_i'^{1/B_i}$, using the bid values *"not willing"* (0.05), *"not entered"* (1), *"in a pinch"* $(1 + 0.5\lambda_{\text{bid}})$, *"willing"* $(1 + 1.5\lambda_{\text{bid}})$, *"eager"* $(1 + 3\lambda_{\text{bid}})$; with $\lambda_{\text{bid}} = 1$. If $S_i < 0.15$ and $K_i$ was not missing, the final score was recomputed as $S_i = \min(K_i^{1/B_i}, 0.15)$. Finally, for reviewers who did not provide their profile for use in conflict-of-interest detection, the final score was reduced by $10\%$.

In all of our analyses, we follow this same procedure to determine the assignment under alternative policies (varying only the parameters $w_{\text{text}}$, $\lambda_{\text{bid}}$, and $q$).

## G   Details Regarding Assumption Suitability

In this section, we provide additional details on the discussion in Section 5 on the suitability of the monotonicity and Lipschitz smoothness assumptions.

**Monotonicity.**   Monotonicity assumes that when any component of the covariates increases, the review quality should not be lower. We can test this assumption on the observed outcomes: among all pairs of reviewer-paper pairs with both outcomes observed, 65.7% (TPDP)/28.0% (AAAI) have a dominance relationship ($X_i \succ X_j$) and of those pairs, 79.8% (TPDP)/76.4% (AAAI) satisfy the monotonicity condition. The fraction of dominant pairs for TPDP is higher since we consider only two covariates.

**Lipschitz Smoothness.**   For the Lipschitz assumption, a choice of distance in covariate space is required. We choose the $\ell_1$ distance, normalized in each dimension so that all component distances lie in $[0, 1]$, and divided by the number of dimensions. For AAAI, some reviewer-paper pairs are missing a covariate; if so, we impute a distance of 1 in that component. We then choose several potential Lipschitz constants $L$ by analyzing the reviewer-paper pairs with observed outcomes.

First, we examine the fraction of pairs of observed reviewer-paper pairs that violate the Lipschitz condition for each value of $L$. Figures 3 and 4 show the CCDF of $L$ for pairs of observations (in other words, the fraction of violating observation pairs for each value of $L$) with respect to expertise and confidence respectively. In our experiments in Section 5, we use values of $L$ corresponding to less than 10%, 5%, and 1% violations from these plots.

Next, we examine the distances from unobserved reviewer-paper pairs to their closest observed reviewer-paper pair. In Figure 5, we show the CCDF of these distances for unobserved reviewer-paper pairs within a set of "relevant" pairs. We define the set of "relevant" unobserved pairs to be all pairs not supported on-policy that have positive probability in at least one policy among all off-policies with varying $w_{\text{text}}$ with $q = 1$ for TPDP, and all off-policies varying $w_{\text{text}}$ and $\lambda_{\text{bid}}$ with $q = 1$ for AAAI.

Intuitively, the Lipschitz assumptions correspond to beliefs that the outcome does not change too much as the similarity components change. As one example, for $L = 30$ on AAAI, when one

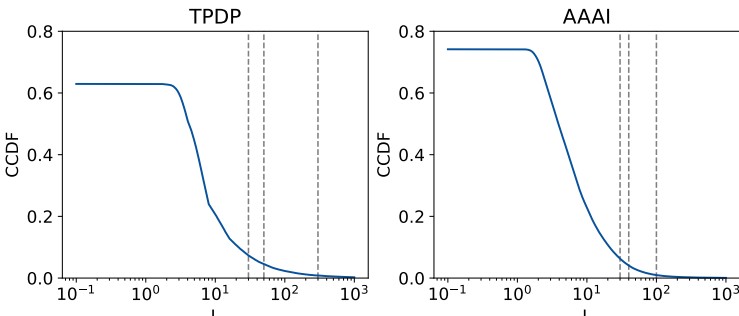

Figure 3: CCDF of the $L = |Y_i - Y_j|/d(X_i, X_j)$ values for all pairs of observed points, where $Y$s are *expertise* scores. The dashed lines denote the $L$ values corresponding to less than $10\%$, $5\%$, and $1\%$ violations. For TPDP, these values are $L = 30, 50, 300$, respectively; for AAAI, $L = 30, 40, 100$.

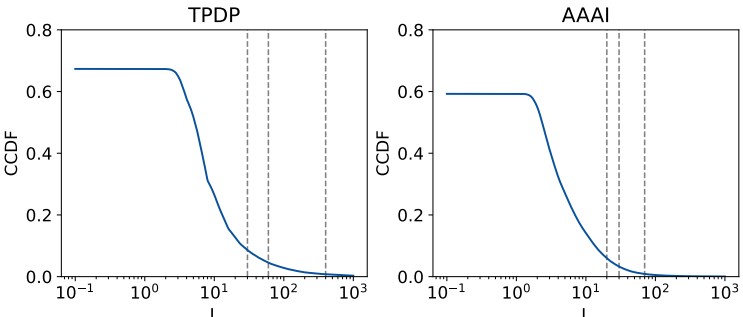

Figure 4: CCDF of the $L = |Y_i - Y_j|/d(X_i, X_j)$ values for all pairs of observed points, where $Y$s are *confidence* scores. The dashed lines denote the $L$ values corresponding to less than $10\%$, $5\%$, and $1\%$ violations. For TPDP, these values are $L = 30, 60, 400$, respectively; for AAAI, $L = 20, 30, 70$.

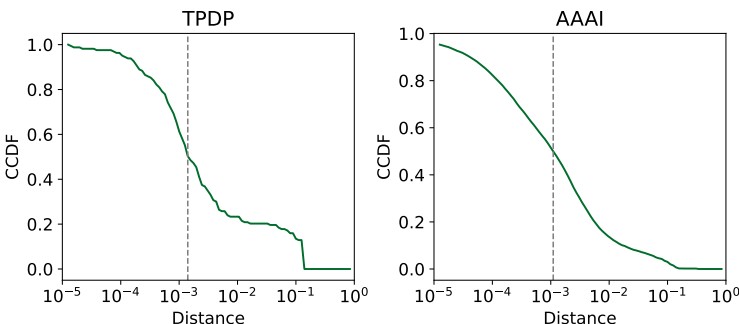

Figure 5: CCDF of the distances between each relevant unobserved reviewer-paper pair and its closest observed reviewer-paper pair. The dashed lines show the medians: $0.0014$ (TPDP) and $0.0011$ (AAAI).

similarity component differs by $0.1$, the outcomes can differ by at most $1$. Effectively, the imputed outcome of each unobserved pair is restricted to be relatively close to the outcome for the closest observed pair. From the distribution in Figure 5, we observe median distances of $0.0014$ (TPDP) and $0.0011$ (AAAI) across the pairs violating positivity under any of these off-policies. We conclude that most imputed pairs are very close to some observed pair, and even large values of $L$ can significantly decrease the bound width when compared to the Manski bounds.

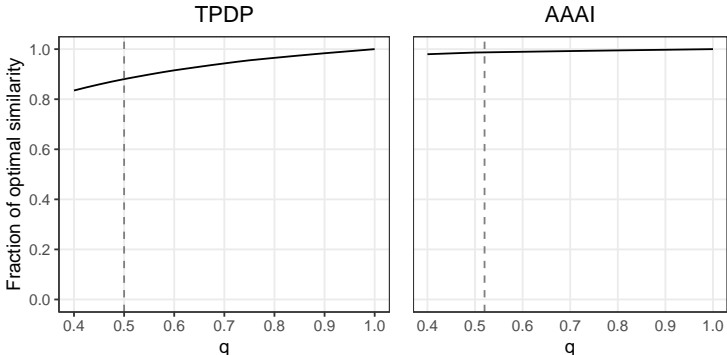

Figure 6: The "cost of randomization" as measured by the expected total assignment similarity. The plot shows the ratio between the sum of similarities under a randomized assignment (LP (3)) with ($q \leq 1$) and the sum of similarities under a deterministic assignment ($q = 1$). The dashed lines show the values of $q$ set on-policy.

## H  Tie-Breaking Behavior

In Section 5 (Datasets), we specify a policy in terms of the parameters of LP (3) (specifically, by altering the parameters $q$, $w_{\text{text}}$, and $\lambda_{\text{bid}}$ from the on-policy values). However, LP (3) may not have a unique solution, and thus each policy may not correspond to a unique set of assignment probabilities. Of particular concern, the on-policy specification of LP (3) does not uniquely identify the actual on-policy assignment probabilities.

Ideally, we could use the same tie-breaking methodology as was used in the on-policy to pick a solution in each off-policy to avoid introducing additional effects from variations in tie-breaking behavior. However, this behavior was not specified in the venues we analyze. To resolve this, we fixed arbitrary tie-breaking behaviors such that the on-policy solution to LP (3) matches the actual on-policy assignment probabilities; we then use these same behaviors for all off-policies.

In the TPDP analyses, we perturb all similarities by small constants such that all similarity values are unique. Specifically, we change the objective of LP (3) to $\sum_{i \in \mathcal{R} \times \mathcal{P}} P(Z_i)[(1 - \lambda)S_i + \lambda \mathcal{E}_i]$, where $\lambda = 1e^{-8}$, and $\mathcal{E} \in \mathbb{R}^{|\mathcal{R}| \times |\mathcal{P}|}$ is the same for all policies. To choose $\mathcal{E}$, we sampled each entry uniformly at random from $[0, 1]$ and checked that the solution of the perturbed on-policy LP matches the on-policy assignment probabilities, resampling until it does. This value of $\mathcal{E}$ was then fixed for all policies.

In the AAAI analyses, the larger size of the similarity matrix meant that randomly choosing an $\mathcal{E}$ that recovers the on-policy solution was not feasible. Instead, we more directly choose how to perturb the similarities in order to achieve consistency with the on-policy. We change the objective of LP (3) to $\sum_{i \in \mathcal{R} \times \mathcal{P}} P(Z_i)(S_i - \epsilon \mathbb{I}[P_A(Z_i) = 0])$, where $\epsilon \in \mathbb{R}$ is chosen for each policy by the following procedure. For each policy, $\epsilon$ is chosen to be the largest value from $\{10^{-9}, 10^{-6}, 10^{-3}\}$ such that the difference in total similarity between the solution of the original and perturbed LPs is no greater than a tolerance of $10^{-5}$. We confirmed that using this procedure to perturb the on-policy LP recovers the on-policy assignment probabilities, as desired.

## I  Similarity Cost of Randomization

In [16], Jecmen et al. empirically analyze the "cost of randomization" in terms of the expected total assignment similarity, i.e., the objective value of LP (3), as $q$ changes. This approach is also used by conference program chairs to choose an acceptable level of $q$ in practice. In Figure 6, we show this trade-off between $q$ and sum-similarity (as a ratio to the optimal deterministic sum-similarity) for both TPDP and AAAI. Note that in contrast, our approach in this work is to measure assignment quality via self-reported expertise or confidence rather than by similarity. In particular, the cost of randomization for TPDP is high in terms of sum-similarity but is revealed by our analysis to be mild in terms of expertise (Section 5, Results).

Table 1: Expertise of bad policies (95% confidence intervals). $L = 50$ for TPDP and $L = 40$ for AAAI.

| Policy | Manski | Monotonicity | Lipschitz |
|--------|--------|--------------|-----------|
| TPDP Max | [2.6115, 2.7045] | [2.6551, 2.6782] | [2.6498, 2.6744] |
| TPDP Min | [2.5521, 2.6126] | [2.5521, 2.5986] | [2.5521, 2.5937] |
| AAAI Max | [3.3919, 3.5213] | [3.4756, 3.4783] | [3.4764, 3.4809] |
| AAAI Min | [3.2591, 3.3846] | [3.3394, 3.3419] | [3.3396, 3.3443] |

## J   Power Investigation: Purposefully Bad Policies

Many of the off-policy assignments we consider in Section 5 have shown to have relatively similar estimated quality. A possible explanation for this tendency is that most "reasonable" optimized policies are roughly equivalent in terms of quality, since our analyses only consider adjusting parameters of the (presumably reasonable) optimized on-policy. To investigate this possibility, we analyze a policy intentionally chosen to have poor quality.

Designing a "bad" policy that can be feasibly analyzed presents a challenge, as the on-policies are both optimized and thus rarely place probability on obviously bad reviewer-paper pairs. To work within this constraint, we look for bad policies where all reviewer pairs with zero on-policy probability are regarded as conflicts. We then contrast the deterministic ($q = 1$) policy that *maximizes* the total similarity score with the "bad" policy that *minimizes* it. Since the on-policy similarities are presumably somewhat indicative of expertise, we expect the minimization policy to be worse.

The results of this comparison are presented in Table 1. On both TPDP and AAAI, we see that our methods clearly identify the minimization policies as worse. The differences in quality between the policies becomes clearer with the addition of Lipschitz and monotonicity assumptions to address attrition. This illustrates that our methods are able to distinguish a good policy (the best of the best matches) from a clearly worse one (the worst of the best matches). Thus, it is likely that our primary analyses are simply exploring high-quality regions of the assignment-policy space, and that peer review assignment quality is often robust to the exact values of the various parameters.

## K   Broader Impact

We hope that our work will have a substantial positive impact on the quality of peer review processes broadly. Our methodology provides conference program chairs and other organizers with a way to evaluate the quality of alternative reviewer-paper assignments that they can easily use for post-mortem analysis. The results from such analyses can be leveraged to adjust the assignment algorithm for future iterations of the conference, ideally leading to improved review quality and improved satisfaction from both reviewers and authors.

We note two potential negative impacts that our work may have. First, our methodology requires making assumptions about the unobserved outcomes (e.g., monotonicity, Lipschitz smoothness) in order to achieve good estimates of assignment quality. If these assumptions do not hold in practice, the resulting estimates may be misleading, potentially supporting adjustments to the assignment policy that in fact decrease review quality. However, as there currently is no principled method for evaluating the quality of alternative assignment policies, the status quo is that such assignment policy choices are made based primarily on the intuition of program chairs. Our work makes the underlying assumptions explicit so that they can be directly considered.

Second, our work considers estimates of the average review quality across all assigned reviewer-paper pairs, analogous to the common sum-of-similarities objective used for review-paper assignment (see Section 2). However, past work has also considered other assignment objectives based on the fairness of the assignment [35, 36, 47, 48]. By focusing only on the average review quality, our analysis does not consider the impact on individual reviewers or papers, potentially encouraging assignment policies that produce higher overall review quality at the cost of fairness. In this work, we focused on average review quality as it is the primary metric currently used by conference organizers, but consider generalizing our approach to estimands based on other notions of fairness a very important direction for future work.

# L   Results for Confidence Outcomes

Figure 7 shows the results of our analyses using *confidence* as a quality measure ($Y$). We find that the results are substantively very similar to those reported in Section 5 using expertise.

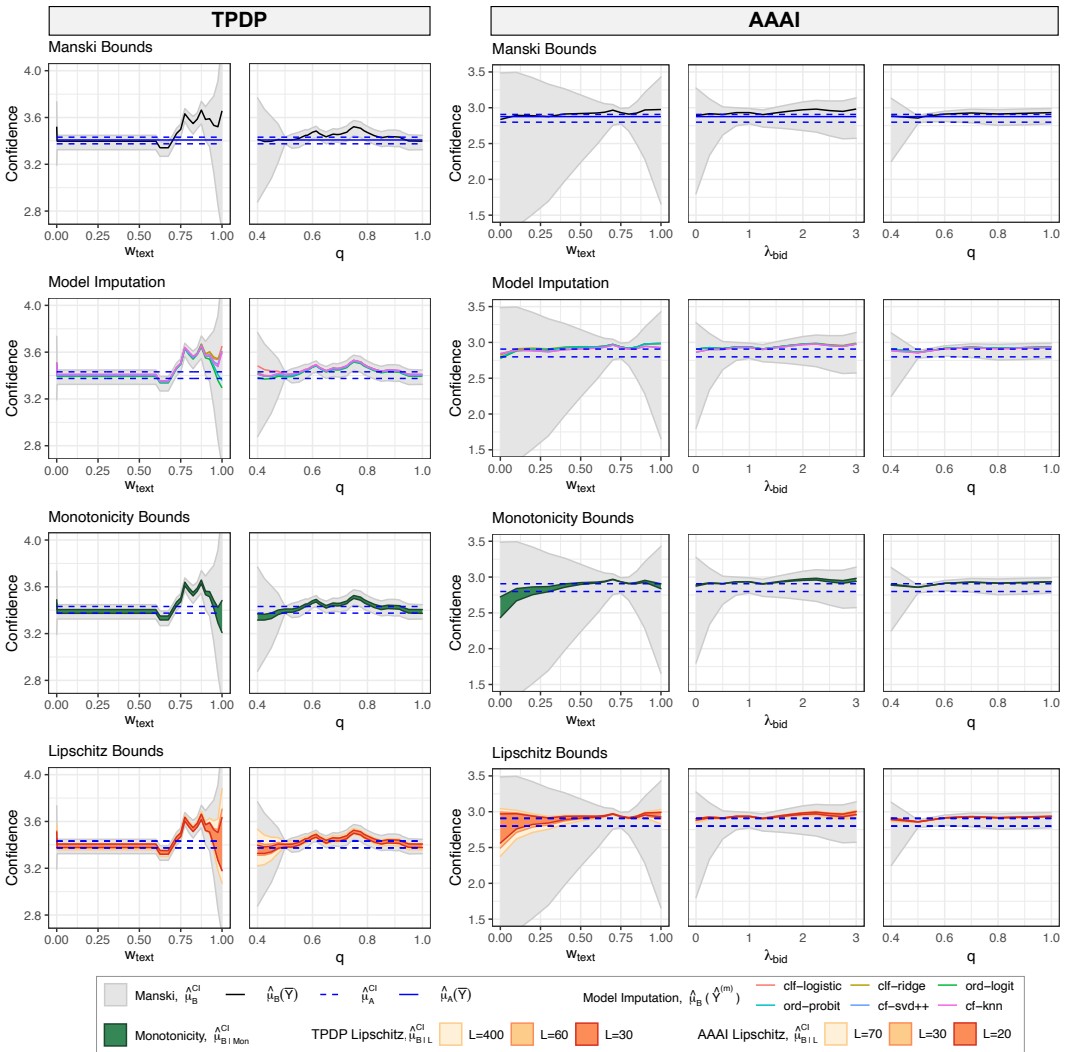

Figure 7: Confidence of off-policies varying $w_{\text{text}}$ and $q$ for TPDP and $w_{\text{text}}$, $\lambda_{\text{bid}}$, and $q$ for AAAI, computed using the different estimation methods described in Section 4. The dashed blue lines indicate Manski bounds around the on-policy confidence, and the grey lines indicate Manski bounds around the off-policy confidence. The error bands (denoted $\widehat{\mu}_B^{CI}$ for the Manski bounds, $\widehat{\mu}_{B|M}^{CI}$ for the monotonicity bounds, and $\widehat{\mu}_{B|L}^{CI}$ for the Lipschitz bounds) represent confidence intervals that asymptotically contain the true value of $\mu_B$ with probability at least $95\%$ as described in Appendix C. Note that to focus on the most relevant regions of the plots, the vertical axes do not start at zero.

