# OpenReview forum: "Counterfactual Evaluation of Peer-Review Assignment Policies"
_NeurIPS.cc/2023/Conference — NeurIPS 2023 spotlight_

### Official Review · Reviewer_5Zed · 2023-07-04

**Soundness:** 2 fair
**Presentation:** 3 good
**Contribution:** 2 fair
**Rating:** 4
**Confidence:** 2

**Summary:**

In this paper, the authors study the counterfactual evaluation of peer-review assignment strategies with randomness. The authors adapt existing off-policy evaluation methods for the specific problem to handle non-positivity, missing reviews and attribution. A framework of off-policy evaluation with different imputation choices are provided. Empirical evaluations are carried out on two datasets to study (1) effect of randomness and (2) weights on different similarity components.

**Strengths:**

1. The paper studied an important problem as online A/B experiments for evaluating peer-review assignments are very expensive and hard to carry out.
2. The authors propose a comprehensive framework based on off-policy evaluation with different choices of imputation methods. Moreover, the authors adapt the existing method to specific cases for paper review.
3. The authors carry out two case studies using the proposed method on two real-world peer review datasets.


**Weaknesses:**

One major concern is that there is no evaluation on the quality for the counterfactual evaluation. It would be much more convincing if some evaluation can be carried out with the help of random A/B experiments or results from existing random experiments. Different methods provide quite different bounds. It is hard to pick the proper method without a ground-truth provided by A/B experiment.

**Questions:**

Please see above section.

**Limitations:**

Yes

---

> ### Author Rebuttal · Authors · 2023-08-09
>
> Thank you for the comments! We appreciate that you recognize the importance of the problem we consider and the value of our off-policy evaluation framework.
>
> > One major concern is that there is no evaluation on the quality for the counterfactual evaluation. It would be much more convincing if some evaluation can be carried out with the help of random A/B experiments or results from existing random experiments. Different methods provide quite different bounds. It is hard to pick the proper method without a ground-truth provided by A/B experiment.
>
> Unfortunately, no existing conference has conducted such an A/B test due to numerous significant costs and challenges. Running an A/B test in a real conference for the purpose of evaluating this work either risks substantially harming the review quality (as some reviewers are assigned under a lower-quality policy) or brings substantial costs in terms of extra reviewer time (if each paper is assigned additional reviewers). Even if we did run an A/B test, if both A and B policies were stochastic we would still likely want to use the off-policy evaluation techniques we develop here to let observations under “B” inform estimates of the “A” policy (and vice versa). Another challenge in executing an A/B test, and considering it as ground truth, is due to issues of interference: we would need to split the pool of reviewers/papers into multiple groups, and each experimental arm would likely influence the results of the other leading to biased estimates. This is a common problem in market experiments and an active research area. Given these challenges, it would be far beyond the scope of this submission to deploy an A/B test in a conference, in addition to the contributions already made in this work.
>
> Our approach in this work is instead to consider the estimates of review quality obtained under a variety of different assumptions, which we verify to be reasonable on the observed outcomes (Appendix G). Where these estimates agree (as they often do), we can draw a conclusion that is likely robust to a violation of any individual assumption; when the estimates disagree, we clearly show how the conclusion depends on which assumption one finds to be most appropriate. We also note that our approach fundamentally relies on randomization in the paper assignment to perform estimation, and thus is not observational in the sense of “observational causal inference”, a context where A/B tests are a relative “gold standard” and highly desired form of validation.

---

> > ### Comment · Reviewer_5Zed · 2023-08-13
> > **Thank you for the rebuttal**
> >
> > Thanks for the response. I have read the rebuttal and remain a borderline 4/5 score. I agree with the authors that it is hard to carry out but still desired as a gold standard to validate the methodology.

---

### Official Review · Reviewer_LpsF · 2023-07-04

**Soundness:** 3 good
**Presentation:** 3 good
**Contribution:** 2 fair
**Rating:** 5
**Confidence:** 3

**Summary:**

This paper leverages recently proposed strategies that introduce randomness in peer-review assignment—in order to mitigate fraud—as a valuable opportunity to evaluate counterfactual assignment strategies.
This paper introduces novel methods for partial identification based on monotonicity and Lipschitz continuity assumptions for the mapping between reviewer-paper covariates and outcomes. The proposed methods are applied to peer-review data from two computer science venues, the authors find that placing higher weight on text similarity results in higher review quality and that introducing randomization in the reviewer-paper assignment only marginally reduces review quality.


**Strengths:**

1. This paper proposes off-policy evaluation as a less costly alternative that exploits existing randomness to enable the comparison of many alternative policies, compared with A/B tests.
2. The paper is well written and organized.
3. The experiments are sufficient and convincing for the conclusion.


**Weaknesses:**

It would be better to compare the proposed off-policy evaluation method with previous commonly used evaluation methods, and analyze the effectiveness of this method.

**Questions:**

1. Why the proposed off-policy evaluation is effective?
2. Compared with other evaluations, can the proposed evaluation method find a better assignment policy?


**Limitations:**

The authors provide the limitations in appendix K.
1. The method requires making assumptions: monotonicity and Lipschitz continuity.
2. The method considers estimates of the average review quality across all assigned reviewer-paper pairs, analogous to the common sum-of-similarities objective used for review-paper assignment, the analysis does not consider the impact on individual reviewers on papers.

---

> ### Author Rebuttal · Authors · 2023-08-09
>
> Thank you for the feedback! We are pleased to hear that you found the paper well-written and the experiments convincing. Below we provide responses to the questions posed in the review.
>
>
> > Why the proposed off-policy evaluation is effective?
>
> The key idea of our proposed approach to off-policy evaluation is that we can exploit already-existing randomness (and data already generated under that randomness) in deployed reviewer-paper assignment designs. This allows us to use causal inference to answer questions about potential alternative paper assignments without intervening in the deployed assignment (which could potentially result in deploying a worse assignment); such questions cannot be credibly answered through analysis of data from deterministic assignments. This work is the first to propose leveraging randomized paper assignments to perform off-policy evaluation in the peer review setting, and our proposed methods are designed to address the challenges of applying standard techniques here (e.g., large numbers of positivity violations).
>
>
> > Compared with other evaluations, can the proposed evaluation method find a better assignment policy?
>
> In our experimental analysis, we do find assignment policies with higher quality than the deployed policies in both venues (Figure 1). In TPDP, increasing $w_{text}$ results in a better assignment; in AAAI, increasing $w_{text}$ and $\lambda_{bid}$ both result in slightly better assignments. We additionally find that randomized policies have a very similar quality to deterministic policies, helping to inform future conference organizers who are considering randomizing their assignments for other reasons (e.g., for the purpose of mitigating fraud). Importantly, the methods we introduce in this work can be used by conference organizers to find the best policy among any class of assignment policies they may have in mind.
>
> We address the comparison with other evaluation methods below in response to the comments in the weaknesses section.
>
> > It would be better to compare the proposed off-policy evaluation method with previous commonly used evaluation methods, and analyze the effectiveness of this method.
>
> As this work is the first to consider off-policy evaluation in this setting, there is only a limited set of baseline methods for comparison. In the causal inference and econometrics literature, the standard approach to estimation in the presence of positivity violations is Manski bounds, which we do consider as one of our methods. Experimentally, we find that the Manski bounds are uninformative on real conference data and are generally unable to distinguish better policies from worse ones. In contrast, our proposed methods result in much sharper estimates by carefully leveraging intuitive assumptions, allowing us to identify policies that improve over the deployed policy.
>
> In practice, the current method for choosing the best assignment policy is highly ad-hoc: conference organizers typically generate a few sample assignments under different policies and spot-check a few of the papers’ assigned reviewers (based on their own prior knowledge). Without the techniques in this work, conference organizers have no way to evaluate an alternative, non-deployed policy in terms of the review quality (e.g., as measured by self-reported expertise). Our approach instead allows for the costless evaluation of any number of alternative policies (for which there is statistical support) based on data from the deployed assignment.

---

### Official Review · Reviewer_4Cne · 2023-07-05

**Soundness:** 3 good
**Presentation:** 3 good
**Contribution:** 3 good
**Rating:** 6
**Confidence:** 3

**Summary:**

The authors consider the problem of evaluating alternative reviewer matching algorithms. They motivate this problem as the cost of running A/B tests. They propose to use off-policy evaluation by exploiting the randomness in review assignments introduced by a fraud-mitigating scheme in recent years. They tackle several main challenges, including positivity and attrition, using imputation, Manski bounds, and improved bounds under monotonicity and continuity assumptions. They apply their method to two CS venues and their findings suggest that randomization does not significantly hurt review quality and higher review quality can be obtained by paying more attention to text similarity.

**Strengths:**

- Sections 1 and 2 are beautifully written.
- The authors have a clear set of challenges that they seek to addresse.
- The authors run comprehensive experiment that yield interesting takeaways.

**Weaknesses:**

- I would suggest stating the assumptions clear and upfront. For example, should state Y_i is not a function of the assignment and no interference assumption.

- The challenges subsection of Section 3 is a little messy compared to the rest of the writing. One way the authors could improve this is by introducing an explicit problem statement. At the moment, it's a mix of discussion, problem statement, and assumptions.

- Section 4 could be improve organizationally. The authors should clearly delineate the proposed approaches & assumptions that go with each.

- The experiment section is strong but could be improved in terms of (i) explanation and (ii) organization of takeaways. For example, the figure is not explained in the text, and it takes some work to understand/parse the notation.


**Questions:**


- The authors consider self reported expertise and confidence as measures of review quality - I could see these as good, but also noisy and limiting proxies. What do the authors expect are the limitations of this approach?

- I'm having trouble telling whether the results given in Section 5 are verified? That is, can we tell whether the proposed approaches improve estimation (beyond the fact that the authors' approaches do what they say they will do, e.g., that the assumptions result in tighter bounds)?

**Limitations:**

The authors discuss a possible limitation in their conclusion. They also discuss their assumptions in-line and refer readers to Appendix G for a discussion on assumption suitability. I do think the authors would benefit from a greater discussion of limitations (particularly if the answer to the second question above is no) and their assumptions in one place.

---

> ### Author Rebuttal · Authors · 2023-08-09
>
> Thank you for the helpful comments! Thank you also for your recommendations on the exposition of the results and the organization of the manuscript. We greatly appreciate them and look forward to improving the clarity of the manuscript with these recommendations in mind.
>
>
> >The authors consider self reported expertise and confidence as measures of review quality - I could see these as good, but also noisy and limiting proxies. What do the authors expect are the limitations of this approach?
>
> We do agree that the self-reported expertise and confidence measures that we use in this work can be noisy proxies for review quality: for example, reviewer self-reported expertise may be miscalibrated with true expertise. Similar concerns motivated us to report the results of both the reviewers’ self-reported expertise (Section 5) and confidence (Appendix L). The fact that the analyses based on both measures of review quality lead to the same substantive conclusions in the two datasets gives us reason to believe that the findings would be robust to other similar outcome measures. We additionally note that our framework and methods are general, and thus can be applied to any outcome/review quality measure available to conference organizers. Such alternative review quality measures could include asking the authors (or meta-reviewers) to rate the quality of the reviews. However, these measures can also be noisy: the authors’ reports may depend on the overall rating of the paper, and the scores of the meta-reviewers may depend on their seniority. Finally, in practice, many venues do not ask meta-reviewers / ACs to score the reviews they oversee, and when they do those scores are often missing, making off-policy evaluation even more challenging. As each measure provides an evaluation of a slightly different aspect of the review process and notion of review quality, the appropriate proxy to use may depend on the goals of the conference organizers.
>
>
> > I'm having trouble telling whether the results given in Section 5 are verified? That is, can we tell whether the proposed approaches improve estimation (beyond the fact that the authors' approaches do what they say they will do, e.g., that the assumptions result in tighter bounds)?
>
> Thank you for encouraging us to think further about verification—the problem setting is such that the ground-truth outcomes are unobserved and cannot be used to directly verify the estimates. Without making assumptions about the outcomes for reviewer-paper pairs violating positivity, the Manski bounds are the only applicable estimates, as they result from imputing minimal and maximal outcomes for these pairs (Section 4). Thus, each of our proposed approaches aims to improve the estimation by leveraging some assumption about these positivity-violating pairs (e.g., the monotonicity and Lipschitz continuity assumptions).
>
> So, while these assumptions again cannot be directly verified on the unobserved outcomes, there are many ways that we indirectly verify that they are reasonable. First, we carefully assess the suitability of these assumptions on the observed outcomes (Appendices G and E). Second, we find experimentally that a diverse set of assumptions lead to converging conclusions: e.g. the estimates of a variety of parametric models tend to lie within the same region as the estimates based on relatively mild Lipschitz/Monotonicity assumptions. This provides some evidence that these estimates are robust to violations of these assumptions; in contrast, estimates that significantly differ under different assumptions should be viewed more skeptically.

---

> > ### Comment · Reviewer_4Cne · 2023-08-10
> > **Thank you for the rebuttal**
> >
> > Thanks for the response. I have read the rebuttal and remain a borderline 5/6 score.
> >
> > As a note: While I understand that it is difficult to evaluate the review setting given the limited metrics available, there are clear limitations to using self-reported expertise and confidence as proxies for review quality. I believe that the authors should clearly acknowledge these limitations in the work and possibly even account for how they could change the findings. I agree that the setting they examine is difficult given data (un)availability and that the framework is general, yet the authors should acknowledge the limitations in the work beyond stating that it is hard to get good data. Similarly, I appreciate that verification is difficult in this setting; please state this and why your analysis is sufficient upfront in Section 5.
> >
> > In general, my main feedback is to be upfront with limitations and assumptions. Doing so makes a work stronger, not weaker. This was a trend in the paper (see my initial review and the paragraph directly above).

---

> > > ### Author Response · Authors · 2023-08-11
> > >
> > > Thank you for the response. We appreciate your feedback and agree that being upfront about the limitations and assumptions of the analyses will make the paper stronger. We attempted to do so in our initial submission within the limited space, and we are happy to revise the manuscript to make the limitations/assumptions more explicit.

---

### Official Review · Reviewer_rhmn · 2023-07-11

**Soundness:** 4 excellent
**Presentation:** 4 excellent
**Contribution:** 3 good
**Rating:** 8
**Confidence:** 3

**Summary:**

This work uses the randomness of a recently-implemented peer review paper assignment algorithm in order to perform off-policy evaluation of other (nearby) randomized assignment strategies. It uses multiple methods for imputing missing values and estimating the errors in the estimators of alternative policies' relative value, including fitting proxy values to the data which obey two plausible structural assumptions. The authors perform this evaluation on paper-reviewer matching data from two conferences, and identify practical and actionable recommendations for modifications to the similarity score calculation and assignment algorithm as currently implemented.

**Strengths:**

This paper has a clear premise, purpose, and scope, and it is well executed.

It introduces structural assumptions into the this off-policy evaluation framework which are technically and theoretically interesting and apparently novel.

It provides actionable recommendations for improving the implementation of paper-reviewer matching algorithms in a major conference, and a framework for performing similar evaluations going forward.

**Weaknesses:**

It seems hard to say how far these recommendations generalize beyond the conferences which are included in the analysis.

It would have been nice to have slightly more explicit discussion of how the various uncertainty estimates are generated in the main body of the work.

It would have been interesting to see this methodology used to compare the current method to a wider range of others---even deterministic similarity-based approaches---which have seen prior use.

**Questions:**

In your discussion of review quality, you mention competing definitions of/proxies for review quality. How does your choice (self-reported expertise and confidence) compare against the other metrics (eg auto-generated and meta-reviewer-evaluated), and would you expect your results to be robust to this choice?

The randomized paper assignment scheme of [17] which you take as the logging policy generates a fractional assignment based on the q constraints and then performs dependent rounding in order to generate an assignment. Since your estimators are linear in these assignments, I presume their means are unaffected by which rounding scheme is used. But what about the variances? (What rounding scheme is used in [17]?) Can you comment on how the choice of rounding scheme might affect the uncertainties as well as your uncertainty bounds, and whether this plays a role in your analysis?

Minor comments (no response expected):
	You might make the definition of review quality which you work with more prominent.
	The choice of T_i for surrogate quality as well as text weight was confusing (unless these were surrogate text values, though that was not my understanding).

**Limitations:**

It would have been great if this work had included more data for analysis, especially from a second large conference. The scope of the recommendations offered seem somewhat limited by conferences which are a part of the analysis. It would be especially interesting to see this analysis performed on multiple years of AAAI data (provided there is a consistent on-policy used) simultaneously.

---

> ### Author Rebuttal · Authors · 2023-08-09
>
> Thank you for the detailed review—we very much appreciate your feedback and are pleased to hear that you found the paper well-executed, the off-policy evaluation framework technically and theoretically interesting, and the recommendations based on our analyses actionable.
>
> > In your discussion of review quality, you mention competing definitions of/proxies for review quality. How does your choice (self-reported expertise and confidence) compare against the other metrics (eg auto-generated and meta-reviewer-evaluated), and would you expect your results to be robust to this choice?
>
> We decided to report the results of both the reviewers’ self-reported expertise (Section 5) and confidence (Appendix L) specifically because we wanted to demonstrate that our substantive results are robust to the particular choice of a review quality metric. The fact that the analyses based on both measures lead to the same substantive conclusions in both datasets gives us reason to believe that the findings would be robust to other related outcome measures as well. We also considered using the scores provided by the meta-reviewers; however, for AAAI 85% of the meta-reviewers’ scores were missing, and for the TPDP workshop these scores were not at all collected by the conference organizers. We note that similar to reviewers’ self-reports, the meta-reviewer scores and other alternative measures, e.g., the authors’ assessment of the reviews, come with some subjective biases as well. For instance, the meta-reviewers’ scores may depend on their seniority, and the authors’ reports depend on the overall rating of the paper (see Section 8.2.2 in [4]). Last, we note that our framework and the methods proposed are general and can be applied to any outcome / review quality measure available to conference organizers.
>
> [4] Nihar B. Shah. Challenges, experiments, and computational solutions in peer review. Communications of the ACM, 65(6):76–87, 2022.
>
>
> >The randomized paper assignment scheme of [17] which you take as the logging policy generates a fractional assignment based on the q constraints and then performs dependent rounding in order to generate an assignment. Since your estimators are linear in these assignments, I presume their means are unaffected by which rounding scheme is used. But what about the variances? (What rounding scheme is used in [17]?) Can you comment on how the choice of rounding scheme might affect the uncertainties as well as your uncertainty bounds, and whether this plays a role in your analysis?
>
> That’s an excellent question that we were also concerned about while developing the analysis framework. Your observation that there is a certain “degree of freedom” in how a deterministic assignment is generated given a fractional assignment is correct. The assignment sampling (or rounding) procedure proposed by Jecmen et al. [17] only guarantees that the marginal probabilities are respected and may choose an arbitrary distribution that does so. The covariance structure of this distribution will affect the variance of our estimators. If the off-policy evaluation was premeditated, it would be an excellent idea to devise an assignment sampling procedure that would proactively minimize the variance and maximize the power of the off-policy evaluation estimates. There are two reasons we opted to use the original procedure proposed by Jecmen et al. [17]. First and foremost, it is the procedure that had already been implemented in OpenReview and other peer-review platforms, and was already used by the conference organizers to generate the pre-existing assignment data we analyzed. Thus, efforts to improve this design would not have been useful to us in this work (but it would be an interesting direction for future work). Second, in practice, we found that accounting for the positivity violations contributes significantly more to the width of the uncertainty intervals than the variance of the estimates in the overlapping regions (where some form of covariance optimization would come in). This observation motivated us to focus our efforts on mitigating the impact of the positivity violations via our proposed estimation methods based on monotonicity and Lipshchtiz continuity assumptions.
>
> [17] Steven Jecmen, Hanrui Zhang, Ryan Liu, Nihar B. Shah, Vincent Conitzer, and Fei Fang. Mitigating manipulation in peer review via randomized reviewer assignments. Advances in Neural Information Processing Systems, 2020.

---

### Decision · Program_Chairs · 2023-09-21

**Decision:**

Accept (spotlight)

**Comment:**

The paper studies how to evaluate peer review assignments using off-policy data (i.e. from previous randomized assignments). All the reviewers agree that the paper is timely (randomized assignments are recently becoming popular at ML conferences), well executed, clearly written, and introduces assumptions and methods that are novel, effective and theoretically interesting. The authors clarified many excellent points raised by the reviewers during the discussion, and these additional details will substantially improve the exposition in a paper revision.